# Decoupling Universal Laws and Environmental Heterogeneity: A Physics-Inspired Framework for Robust Spatio-Temporal Forecasting

Aoyu Liu [1]  Liming Wei [1]  Yaying Zhang [1]

## Abstract

Most spatio-temporal forecasting models assume in-distribution data and can degrade sharply under non-stationary environments. Existing methods for handling distribution shift largely rely on discrete graph inference, making it difficult to disentangle universal dynamics from environment-specific changes and to respect the continuous physical nature of spatio-temporal fields. To this end, we propose STPDE, a general framework that reformulates spatio-temporal dynamics as the evolution of inhomogeneous partial differential equations. STPDE explicitly decomposes dynamics into an *Invariant Diffusion Operator* that captures universal mechanisms and an *Environment Basis Manifold* that parameterizes local heterogeneous media. We show that the Green's function of the Laplacian can be effectively approximated by linear attention, enabling global diffusion at scale. Combined with stochastic environment perturbations, STPDE improves robustness under heterogeneous and shifting environments. Extensive experiments on in-distribution forecasting, out-of-distribution generalization, few-shot cross-city transfer, and continual learning demonstrate consistent improvements over state-of-the-art baselines with competitive computational efficiency.

## 1. Introduction

Spatio-temporal forecasting is a core problem in intelligent transportation systems and smart cities (Yang et al., 2025; Liu et al., 2025b). It seeks to model a continuously evolving physical field where temporal dynamics are tightly coupled with spatial interactions. Although deep learning methods

[1]The Key Laboratory of Embedded System and Service Computing, Ministry of Education, Tongji University, Shanghai, China. First author email: liuaoyu@tongji.edu.cn. Correspondence to: Yaying Zhang <yaying.zhang@tongji.edu.cn>.

*Proceedings of the 43rd International Conference on Machine Learning*, Seoul, South Korea. PMLR 306, 2026. Copyright 2026 by the author(s).

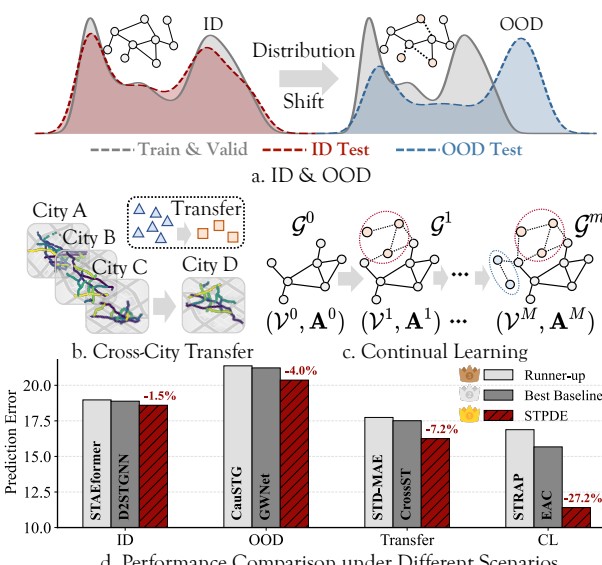

*Figure 1.* Background of robust spatio-temporal forecasting.

have shown strong performance in controlled settings (Kumar et al., 2024; Liang et al., 2025; Liu et al., 2025d), real-world deployment remains difficult due to pervasive distribution shifts. As shown in Figure 1, models must not only achieve high accuracy under the in-distribution (ID) regime, but also remain robust to non-stationary environments induced by cross-city transfer, urban evolution, seasonal variation, and unexpected events. Building spatio-temporal forecasters that are reliable under these conditions remains an open challenge.

Most existing spatio-temporal graph neural networks (STGNNs) (Li et al., 2018; Wu et al., 2019) are optimized for a single training distribution. They rely on fixed or learned graph structures that absorb dataset-specific topological biases. Effective on standard benchmarks, they fail under distribution shift, as seen in cross-city transfer or incremental deployment settings. To address this, researchers have explored adaptation strategies like pre-training followed by fine-tuning (Li et al., 2024; Liu & Zhang, 2025) and continual learning (Chen & Liang, 2025; Wang et al., 2023b). However, these methods largely re-fit statistical regularities in new environments, rather than endowing the

model with an intrinsic mechanism that preserves ID accuracy and improves out-of-distribution (OOD) generalization.

Recent studies (Wang et al., 2025) have begun to target spatio-temporal OOD generalization by incorporating causal discovery, invariant representation learning (Zhou et al., 2023; Xia et al., 2023), and prompting-based techniques (Wang et al., 2024; Ma et al., 2025b). While these methods can improve robustness under certain shifts, they often cast spatio-temporal evolution as a discrete graph inference problem, overlooking the intrinsically continuous physical nature of spatio-temporal fields. More critically, existing approaches have limited ability to disentangle universal evolution mechanisms from environment-specific variations. Consequently, models are frequently forced into a trade-off between exploiting environmental cues to boost in-distribution (ID) performance and suppressing them to enhance generalization—a tension that becomes especially pronounced in highly non-stationary environments.

To address these limitations, we propose STPDE, a physics-inspired framework that reformulates spatio-temporal forecasting as the numerical evolution of an inhomogeneous partial differential equation (PDE). Under this view, spatio-temporal observations are treated as samples from an underlying latent continuous field whose dynamics are governed by an inhomogeneous PDE. Universal mechanisms are captured by invariant differential operators (e.g., the Laplacian), which are shared across environments and thus support transferability and continual evolution. In contrast, spatially varying coefficients and source terms explicitly encode environment-dependent properties, enabling flexible modeling of spatial heterogeneity.

Building on this perspective, STPDE integrates three key components. First, we design an *Invariant Diffusion Operator* that approximates the Green's function of the governing PDE via linear graph attention, enabling global diffusion modeling with linear complexity. Second, we introduce an *Environment Basis Manifold* that parameterizes spatially varying coefficients through a sparse basis representation with adaptive routing, allowing the model to fit local environments while remaining amenable to efficient adaptation under distribution shifts. Third, we inject *stochastic perturbations* into the basis manifold to emulate unobserved environmental changes, encouraging the invariant operator to capture more transferable dynamics. By explicitly decoupling invariant dynamics from environment-specific modulation, STPDE offers a unified approach to spatio-temporal forecasting under distribution shift.

Our main contributions are as follows: ❶ We propose STPDE, a PDE-driven framework that decouples invariant evolution mechanisms from environment-dependent heterogeneity, providing a unified formulation for robust spatio-temporal forecasting under distribution shift. ❷ We pro-

pose an *Invariant Diffusion Operator* that captures domain-invariant dynamics and enables efficient global propagation modeling with linear complexity. ❸ We introduce an *Environment Basis Manifold* with adaptive parameterization and stochastic perturbations, which flexibly captures local heterogeneity while improving robustness in highly non-stationary environments. ❹ We conduct extensive experiments on ID forecasting, OOD generalization, few-shot cross-city transfer, and continual learning, demonstrating that STPDE consistently outperforms state-of-the-art baselines in accuracy, robustness, and efficiency.

## 2. Related Work

STGNNs dominate spatio-temporal forecasting by jointly modeling spatial and temporal dependencies. Foundational models such as STGCN (Yu et al., 2018) and GWNet (Wu et al., 2019) combine graph and temporal convolutions, with GWNet further learning adaptive matrices. Later work highlights that strong representation design can rival complex architectures: STID (Shao et al., 2022a) shows competitive performance with simple MLPs and identity embeddings, while STAEformer (Liu et al., 2023a) injects adaptive embeddings into Transformers. However, STGNNs often degrade under distribution shifts; large-scale benchmarks confirm substantial performance drops under seasonal and cross-year shifts (Wang et al., 2025). To improve OOD generalization, causal invariant learning methods (e.g., CauSTG (Zhou et al., 2023), CaST (Xia et al., 2023)) disentangle invariant relations from environment-specific factors, and robustness-oriented designs such as STOP (Ma et al., 2025b) further enhance stability. Beyond supervised training, spatio-temporal pre-training via masked modeling (STEP (Shao et al., 2022b), GPT-ST (Li et al., 2023)) learns transferable representations, while prompt-based adaptation (FlashST (Li et al., 2024)) improves cross-domain efficiency. For streaming settings, continual learning methods like TrafficStream (Chen et al., 2021) and prompt-based EAC (Chen & Liang, 2025) adapt to evolving patterns while mitigating forgetting.

## 3. Preliminaries

**Problem Definition.** Let $\mathcal{G} = (\mathcal{V}, \mathbf{A})$ be a spatio-temporal graph with $N = |\mathcal{V}|$ nodes, where $\mathcal{V}$ is the node set and $\mathbf{A} \in \mathbb{R}^{N \times N}$ is the adjacency matrix. At time step $t$, node features over the graph are represented by a signal $\mathbf{X}_t \in \mathbb{R}^{N \times C}$, where $C$ is the number of feature channels. The goal of *spatio-temporal forecasting* is to learn a parameterized mapping $\mathcal{F}_\theta$ that predicts future states from historical observations. Formally, given a historical window of length $T$, $\mathcal{X} = [\mathbf{X}_{t-T+1}, \dots, \mathbf{X}_t] \in \mathbb{R}^{T \times N \times C}$, the model predicts the next $T'$ steps $\mathcal{Y} = [\mathbf{X}_{t+1}, \dots, \mathbf{X}_{t+T'}] \in \mathbb{R}^{T' \times N \times C}$.

**Task Settings.** We evaluate spatio-temporal forecasting under four representative settings that span both stationary and non-stationary scenarios: ❶ *in-distribution (ID) forecasting*, where training and test data are sampled from the same stationary distribution, i.e., $P_{\text{train}} = P_{\text{test}} = \mathcal{P}$; ❷ *out-of-distribution (OOD) generalization*, where non-stationary factors induce a distribution shift ($P_{\text{train}} \neq P_{\text{test}}$) and models must generalize across a family of environments $\mathcal{E}$; ❸ *few-shot cross-city transfer*, which adapts from a source domain $\mathcal{D}_{\text{src}}$ to a data-scarce target domain $\mathcal{D}_{\text{tgt}}$ with disjoint node sets $\mathcal{V}_{\text{src}} \cap \mathcal{V}_{\text{tgt}} = \emptyset$ (and potentially different graph structures); ❹ *continual learning on evolving graphs*, where data arrive as a task stream $\{\mathcal{G}^0, \mathcal{G}^1, \ldots, \mathcal{G}^M\}$ and the node set expands over time as $\mathcal{V}^{(m)} = \mathcal{V}^{(m-1)} \cup \Delta\mathcal{V}^{(m)}$, requiring models to incorporate newly introduced nodes while retaining performance on previously learned ones.

## 4. Methodology

### 4.1. The Theory–Reality Bridge

Motivated by recent advances in physics-inspired modeling (Wu et al., 2023; 2025b; Ji et al., 2022), we recast spatio-temporal dynamics from a continuous field-theoretic perspective, moving beyond discrete message passing. As shown in Figure 2, the *Theory–Reality Bridge* maps the real-world *Scenario Space* (e.g., time-stamped traffic states on a road network) into a *Theory Space* of latent continuous fields and their governing propagation operators.

**Theory.** A classical heat model assumes isotropic, spatially homogeneous diffusion, $\frac{\partial \mathbf{u}}{\partial \tau} = \bar{\alpha} \nabla^2 \mathbf{u} + \bar{\mathcal{S}}$. While analytically convenient, it cannot reflect ubiquitous *spatial heterogeneity* (region-dependent media) or *non-stationarity* (time-varying exogenous perturbations). We therefore model the latent field $\mathbf{u}(\mathbf{x}, \tau)$ as an *inhomogeneous diffusion* process driven by stochastic environmental bases:

$$\frac{\partial \mathbf{u}(\mathbf{x}, \tau)}{\partial \tau} = \alpha(\Phi, \xi) \nabla^2 \mathbf{u}(\mathbf{x}, \tau) + \mathcal{S}(\Phi, \xi), \quad (1)$$

where $\tau$ denotes continuous physical time, distinct from the discrete observation index $t$. Eq. (1) factorizes the dynamics into: ❶ a shared, domain-invariant Laplacian $\nabla^2$ that captures topology-aware transport and conservation; ❷ an environment-conditioned diffusivity $\alpha(\Phi, \xi)$ defined on the *Environment Basis Manifold* $\Phi$ (with latent perturbations $\xi$) to adapt propagation strength to local conditions; and ❸ an environment-aware source $\mathcal{S}(\Phi, \xi)$ that models non-diffusive exogenous drivers (e.g., holidays and incidents), enabling temporal drift.

**Model Implementation.** As shown in Figure 2, STPDE enables efficient and robust rollouts for large-scale spatio-temporal forecasting. Unlike step-wise autoregressive recurrence (e.g., RNNs) (Bai et al., 2020; Shao et al., 2022c), we adopt a *time-as-features* principle: temporal dependencies

are compressed into latent features, and future evolution is produced by a single-shot solver. Concretely, STPDE instantiates a neural numerical solver for Eq. (1), mapping history $\mathcal{X}$ directly to forecasts $\hat{\mathcal{Y}}$ in three stages. First, a linear projection $\phi_{\text{in}}$ lifts and compresses the length-$T$ history into a $D$-dimensional latent field on nodes: $\mathcal{X} \in \mathbb{R}^{T \times N \times C} \xrightarrow{\phi_{\text{in}}} \mathbf{h} \in \mathbb{R}^{N \times D}$. This removes recurrent dependencies along time and enables parallel inference. Then, the physics-inspired PDE solver evolves $\mathbf{h}$ in a single forward pass by simulating Eq. (1). It infers node-wise diffusivity $\alpha(\Phi, \xi)$ and source $\mathcal{S}(\Phi, \xi)$ from the *Environment Basis Manifold*, and uses them to modulate a shared invariant Laplacian $\nabla^2$. This factorization preserves universal transport and conservation structure while adapting to spatial heterogeneity and non-stationary perturbations. Finally, to absorb high-frequency residuals induced by numerical approximation and to improve expressiveness under drift, we append an FFN with SwiGLU after PDE solving. The evolved state $\tilde{\mathbf{h}} \in \mathbb{R}^{N \times D}$ is mapped back via $\phi_{\text{out}}$ to produce $\hat{\mathcal{Y}} \in \mathbb{R}^{T' \times N}$. Overall, the resulting *Encode–Solve–Decode* pipeline is end-to-end trainable, efficient, and physically grounded.

### 4.2. The Invariant Diffusion Operator

In STPDE, the *Invariant Diffusion Operator*—the Laplacian $\nabla^2$—serves as the mechanistic backbone governing physical evolution. Many real-world spatio-temporal systems (e.g., urban traffic networks) are characterized by pronounced *long-range* dependencies, in which localized perturbations can propagate through the underlying medium and induce coherent, system-wide responses. Motivated by classical partial differential equation (PDE) theory (Wu et al., 2025a), we interpret the evolution of a spatio-temporal field as a kernel integral transform induced by Green's functions, and further reveal its structural correspondence to global (linear) attention mechanisms.

Consider the diffusion equation $\partial\mathbf{u}(\mathbf{x}, \tau)/\partial\tau = \nabla^2\mathbf{u}(\mathbf{x}, \tau)$, posed on a spatial domain $\Omega$ with appropriate boundary conditions. Its solution admits the Green's function (heat kernel) representation:

$$\mathbf{u}(\mathbf{x}, \tau) = \int_\Omega \mathcal{K}(\mathbf{x}, \mathbf{x}'; \tau)\, \mathbf{u}(\mathbf{x}', 0)\, d\mathbf{x}'. \quad (2)$$

Eq. (2) yields a clear physical interpretation: the state at location $\mathbf{x}$ and diffusion time $\tau$ is a superposition of contributions from *all* source locations $\mathbf{x}'$, weighted by the heat kernel $\mathcal{K}(\mathbf{x}, \mathbf{x}'; \tau)$, which characterizes how signals diffuse through the medium. This integral formulation induces a form of *global* coupling over the domain, closely mirroring the structure of attention mechanisms, where each query location aggregates information from all keys via a kernel-defined affinity.

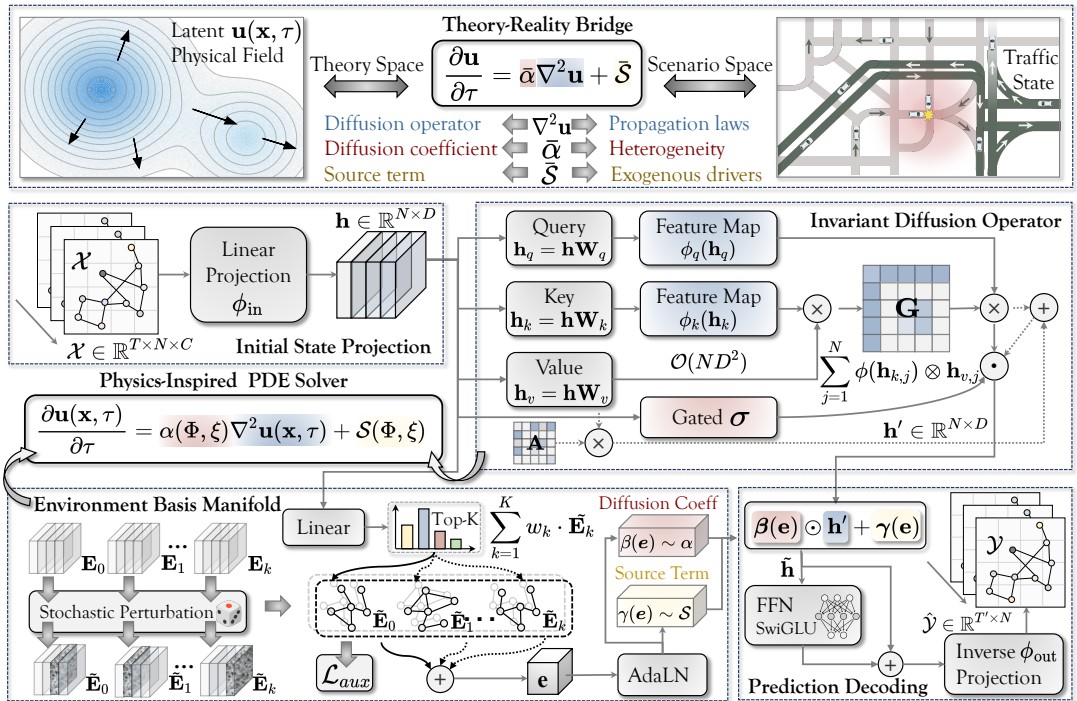

*Figure 2.* The conceptual framework of STPDE: Bridging continuous field theory with a discrete physics-inspired neural architecture.

**From Kernel Propagation to Global Attention.** We encode the observed spatio-temporal field into latent node states $\mathbf{h} \in \mathbb{R}^{N \times D}$, where the $i$-th node embedding $\mathbf{h}_i \in \mathbb{R}^D$ is treated as a row vector. We apply learnable linear projections to obtain the query, key, and value representations:

$$\mathbf{h}_{q,i} = \mathbf{h}_i \mathbf{W}_q, \quad \mathbf{h}_{k,i} = \mathbf{h}_i \mathbf{W}_k, \quad \mathbf{h}_{v,i} = \mathbf{h}_i \mathbf{W}_v, \quad (3)$$

where $\mathbf{W}_q, \mathbf{W}_k, \mathbf{W}_v \in \mathbb{R}^{D \times D}$ are learnable parameters. By parameterizing the discrete propagation kernel over node pairs $(i, j)$ using normalized inner products of the projected features, the resulting update reduces to the standard form of global attention:

$$\mathbf{h}_i' = \sum_{j=1}^{N} \frac{\exp(\mathbf{h}_{q,i} \mathbf{h}_{k,j}^\top / \sqrt{D})}{\sum_{z=1}^{N} \exp(\mathbf{h}_{q,i} \mathbf{h}_{k,z}^\top / \sqrt{D})} \mathbf{h}_{v,j}. \quad (4)$$

This perspective makes the correspondence explicit: kernel-based physical propagation aggregates information globally, while attention implements such aggregation by learning an affinity kernel directly from data.

**Theorem 4.1.** *Let $\mathcal{G} = (\mathcal{V}, \mathbf{A})$ be a graph with $N$ nodes, and let $\mathcal{K}(\tau) \in \mathbb{R}^{N \times N}$ denote the discrete heat kernel at diffusion time $\tau$. Assume that $\mathcal{K}(\tau)$ can be approximated, up to row-wise normalization, by a feature-induced kernel $\hat{\mathcal{K}}_{ij}(\tau) \propto \exp(\mathbf{h}_{q,i}^\top \mathbf{h}_{k,j})$. Then the Green's-function propagation $\mathbf{h}' = \mathcal{K}(\tau)\mathbf{h}_v$ is approximated by the global attention update in Eq. (4), where the row-normalized attention weights realize diffusion-type coupling over all nodes.*

**Linear-Complexity Diffusion via Kernel Factorization.** The $\mathcal{O}(N^2)$ computational cost of Eq. (4) becomes prohibitive for large graphs. To address this limitation, we adopt a separable kernel approximation of the form:

$$\mathcal{K}_{ij} \approx \phi_q(\mathbf{h}_{q,i})^\top \phi_k(\mathbf{h}_{k,j}), \quad (5)$$

where $\phi_q(\cdot)$ and $\phi_k(\cdot)$ are feature maps (we use a softmax-based feature map). Exploiting associativity, the computation can be reorganized into an "aggregate–distribute" procedure by first constructing a global physical context matrix:

$$\mathbf{G} = \sum_{j=1}^{N} \phi_k(\mathbf{h}_{k,j}) \mathbf{h}_{v,j}^\top \in \mathbb{R}^{D \times D}, \quad (6)$$

and then retrieving node-wise diffusion components via:

$$\mathbf{h}_i' = \mathbf{G}^\top \phi_q(\mathbf{h}_{q,i}). \quad (7)$$

By avoiding the explicit computation of the $N \times N$ affinity matrix, this method reduces complexity to $\mathcal{O}(ND^2)$, achieving linearity in $N$ for a fixed dimension $D$.

**Theorem 4.2.** *Assume the kernel admits the separable form in Eq. (5), and define $\mathbf{G}$ as in Eq. (6). Then the global kernel propagation $\mathbf{h}_i' = \sum_{j=1}^{N} \phi_q(\mathbf{h}_{q,i})^\top \phi_k(\mathbf{h}_{k,j}) \mathbf{h}_{v,j}$ can be computed exactly by the two-step procedure in Eqs. (6)–(7), thereby reducing the attention-style global coupling from $\mathcal{O}(N^2)$ to $\mathcal{O}(ND^2)$ time.*

The proofs of Theorem 4.1 and Theorem 4.2 are provided in Section B and C.

**Optional Local Geometric Prior.** To exploit domain locality, we optionally inject a predefined adjacency $\mathbf{A} \in \mathbb{R}^{N \times N}$ (e.g., road connectivity). We augment the global diffusion update with a parallel local aggregation:

$$\mathbf{h}'_i = \mathbf{G}^\top \phi_q(\mathbf{h}_{q,i}) + \sum_{j=1}^N \mathbf{A}_{ij} \mathbf{h}_{v,j}. \qquad (8)$$

This provides an explicit trade-off between physical generalization and task-specific fidelity: in stationary ID regimes, $\mathbf{A}$ can speed up convergence and sharpen local details, while under OOD shifts a fixed graph prior may be restrictive. We therefore use $\mathbf{A}$ as an optional inductive bias.

### 4.3. Environment Basis Manifold

Although the *Invariant Diffusion Operator* encodes universal transport structure, real-world dynamics are heavily shaped by *heterogeneity* and *non-stationarity*. We capture these effects via an *Environment Basis Manifold* $\Phi$ and stochastic perturbations $\boldsymbol{\xi}$: $\Phi$ provides a learnable, structured environment representation, while $\boldsymbol{\xi}$ induces local random shifts. Together, $(\Phi, \boldsymbol{\xi})$ parameterizes environment-conditioned coefficients (e.g., $\alpha(\Phi, \boldsymbol{\xi}), \mathcal{S}(\Phi, \boldsymbol{\xi})$). Inspired by topology-free spatial embeddings (Shao et al., 2022a; Dong et al., 2024), we reconstruct node-wise environment embeddings by sparse routing over $\Phi$, enabling adaptive propagation and disturbance modeling without per-node free parameters.

Specifically, we represent environments using $K$ shared bases $\Phi = \{\mathbf{E}_k\}_{k=1}^K \in \mathbb{R}^{K \times N \times D}$, where $\mathbf{E}_k[i, :] \in \mathbb{R}^D$ denotes the node embedding under basis $k$. This compact basis set captures diverse conditions while encouraging reuse across nodes, improving robustness under distribution shift.

**Stochastic Perturbation.** To prevent overfitting, we perturb the bases when forming node-wise embeddings: for each node $i$, we construct perturbed instances $\tilde{\mathbf{E}}_k^{(i)}$ by *real–average–misaligned* sampling—selecting ❶ the original basis $\mathbf{E}_k$, ❷ an expert-specific "average" embedding (implemented as a learnable shared vector), or ❸ a permuted basis $\Pi(\mathbf{E}_k)$ that swaps node indices to inject structured misalignment. This stochastic construction expands the neighborhood induced by $\boldsymbol{\xi}$ during training and improves robustness for OOD generalization and cross-city transfer.

**Sparse Routing and Coefficient Modulation.** Conditioned on the latent state $\mathbf{h}_i \in \mathbb{R}^D$, a router $\mathcal{R} : \mathbb{R}^D \to \mathbb{R}^K$ outputs logits over the bases, from which we compute sparse Top-$k$ mixture weights:

$$w_{i,k} = \frac{\exp([\mathcal{R}(\mathbf{h}_i)]_k)}{\sum_{j \in \text{Top-}k(i)} \exp([\mathcal{R}(\mathbf{h}_i)]_j)} \cdot \mathbb{I}(k \in \text{Top-}k(i)), \qquad (9)$$

where Top-$k(i)$ denotes the indices of the $k$ largest logits

for node $i$. We reconstruct a node-wise embedding:

$$\mathbf{e}_i = \sum_{k=1}^K w_{i,k} \tilde{\mathbf{E}}_k^{(i)}[i, :] \in \mathbb{R}^D, \qquad (10)$$

which serves as a discrete proxy of the local environment. To couple $\mathbf{e}_i$ with the continuous diffusion solver, we adopt Adaptive Layer Normalization (AdaLN) to modulate the output $\mathbf{h}'_i$ of the *Invariant Diffusion Operator*:

$$\tilde{\mathbf{h}}_i = \boldsymbol{\gamma}(\mathbf{e}_i) \odot \mathbf{h}'_i + \boldsymbol{\beta}(\mathbf{e}_i), \qquad (11)$$

where $\boldsymbol{\gamma}(\cdot)$ and $\boldsymbol{\beta}(\cdot)$ are learned mappings $\mathbb{R}^D \to \mathbb{R}^D$.

**Load Balancing.** To encourage diverse utilization of all bases in $\Phi$, we introduce a load-balancing loss. Let $P_k$ denote the marginal usage of basis $k$ over a mini-batch $B$:

$$P_k = \frac{1}{|B| N} \sum_{x \in B} \sum_{i=1}^N w_{i,k}. \qquad (12)$$

We minimize:

$$\mathcal{L}_{\text{aux}} = K \sum_{k=1}^K P_k^2 - 1, \mathcal{L}_{\text{total}} = \mathcal{L}_{\text{task}} + \lambda \mathcal{L}_{\text{aux}}, \qquad (13)$$

which is minimized when $P_k = 1/K$ for all $k$, thereby mitigating router collapse, promoting diverse basis activation, and stabilizing environment-conditioned generalization.

## 5. Experiments

### 5.1. Experimental Setting

**Datasets and Protocols.** We evaluate STPDE on two widely-used traffic forecasting benchmarks, **LargeST** (Liu et al., 2023b) and **PEMS-Stream** (Chen et al., 2021), under four settings. For *ID forecasting*, we use four LargeST sub-regions: **CA-D3** (480 nodes), **SD** (716 nodes), **Orange** (953 nodes), and **LA** (1,728 nodes); we train and validate on January 2019 with a 3:1 split and test on *a subset* of February 2019. For *OOD generalization*, we keep the same January 2019 training protocol but evaluate on *a subset* of August 2019 to assess seasonal distribution shifts. For *few-shot cross-city transfer*, we pre-train on the first two weeks of December 2018 from **SD**, **Orange**, and **LA**, and then transfer to **CA-D3**, where only the first 10% of the target-domain training data is available for adaptation. For *continual learning*, we adopt the streaming benchmark PEMS-Stream (Chen et al., 2021), which contains seven incremental periods spanning 2011–2017 with a dynamically growing node set (655→871 nodes), to evaluate incremental spatio-temporal modeling under evolving topologies.

**Baselines.** We compare STPDE with baselines from four categories. For STGNNs, we include **GWNet** (Wu et al.,

*Table 1.* Performance comparison: ID forecasting vs. OOD generalization. **Red Bold** indicates best, Blue Underline indicates second best.

| Dataset | Type | Metric | GWNet | STID | D²STGNN | STAEF | HimNet | STPGNN | PatchSTG | BiST | CauSTG | STONE | STOP | STPDE |
|---|---|---|---|---|---|---|---|---|---|---|---|---|---|
| **CA-D3** | ID | MAE | 14.66 | 14.89 | 14.86 | 14.99 | 16.17 | 15.65 | 15.93 | 15.61 | 16.07 | 17.51 | 15.26 | **13.57** |
| | | RMSE | 24.49 | 25.92 | 25.28 | 25.01 | 29.00 | 26.21 | 27.21 | 27.59 | 26.18 | 28.86 | 26.15 | **23.08** |
| | | MAPE (%) | 11.88 | 12.22 | 11.84 | 12.69 | 12.63 | 12.85 | 12.47 | 12.14 | 12.82 | 15.59 | 12.45 | **11.45** |
| | OOD | MAE | 16.84 | 18.61 | 17.80 | 17.82 | 19.37 | 17.73 | 18.98 | 19.61 | 17.20 | 17.89 | 17.68 | **16.34** |
| | | RMSE | 25.00 | 27.45 | 26.49 | 26.15 | 29.10 | 26.32 | 28.41 | 29.54 | 25.19 | 26.17 | 26.18 | **24.12** |
| | | MAPE (%) | 12.44 | 14.78 | 13.14 | 14.29 | 15.80 | 13.50 | 14.20 | 13.99 | 13.64 | 17.29 | 14.04 | **12.02** |
| **SD** | ID | MAE | 19.24 | 18.84 | 18.92 | 18.76 | **18.54** | 21.75 | 18.89 | 18.64 | 21.33 | 23.52 | 22.85 | 18.96 |
| | | RMSE | 30.18 | 29.86 | 29.80 | 29.94 | 29.89 | 34.10 | 30.33 | **29.72** | 33.41 | 36.93 | 35.56 | 29.99 |
| | | MAPE (%) | 12.08 | 11.68 | 11.94 | 11.50 | **11.41** | 13.99 | 11.72 | 11.56 | 13.34 | 17.07 | 14.36 | 12.01 |
| | OOD | MAE | 20.58 | 22.77 | 21.68 | 22.20 | 22.15 | 22.31 | 22.06 | 22.09 | 21.25 | 22.26 | 23.71 | **19.89** |
| | | RMSE | 31.28 | 34.74 | 33.24 | 33.84 | 34.90 | 33.53 | 33.65 | 35.14 | 31.91 | 33.41 | 35.63 | **30.21** |
| | | MAPE (%) | 12.70 | 13.77 | 12.58 | 14.27 | 14.03 | 13.32 | 12.74 | 16.98 | 12.98 | 14.43 | 14.43 | **11.43** |
| **Orange** | ID | MAE | 20.23 | 19.75 | **19.25** | 19.47 | 19.36 | 21.70 | 19.76 | 19.48 | 22.29 | 25.69 | 23.06 | 19.53 |
| | | RMSE | 32.17 | 32.26 | **31.07** | 31.93 | 32.07 | 35.01 | 32.47 | 32.01 | 35.26 | 40.21 | 36.74 | 31.63 |
| | | MAPE (%) | 17.80 | 17.18 | 17.46 | **16.27** | 16.43 | 18.16 | 17.18 | 17.34 | 19.54 | 30.76 | 19.95 | 16.34 |
| | OOD | MAE | 22.22 | 24.66 | 23.16 | 24.16 | 23.71 | 23.17 | 23.71 | 23.49 | 22.02 | 24.33 | 23.96 | **21.52** |
| | | RMSE | 34.19 | 37.73 | 35.56 | 37.51 | 37.44 | 35.88 | 36.95 | 36.51 | 33.79 | 36.98 | 36.63 | **33.28** |
| | | MAPE (%) | **18.06** | 25.65 | 19.48 | 19.56 | 21.13 | 18.99 | 19.51 | 20.19 | 18.45 | 24.41 | 19.43 | 18.88 |
| **LA** | ID | MAE | 24.52 | 23.24 | 22.50 | 22.67 | 22.97 | 24.42 | 23.52 | 22.81 | 26.06 | 29.59 | 25.73 | **22.33** |
| | | RMSE | 38.15 | 37.14 | 35.71 | 36.55 | 37.43 | 38.11 | 37.91 | 36.98 | 40.07 | 44.66 | 40.36 | **35.55** |
| | | MAPE (%) | 10.45 | 9.82 | 9.40 | 9.65 | 9.82 | 10.17 | 9.90 | 9.56 | 11.22 | 15.29 | 10.76 | **9.31** |
| | OOD | MAE | 25.25 | 27.85 | 26.01 | 26.30 | 27.06 | 25.92 | 27.42 | 26.80 | 25.00 | 27.27 | 25.57 | **23.71** |
| | | RMSE | 39.07 | 42.70 | 40.18 | 40.08 | 43.40 | 39.83 | 42.66 | 42.40 | 38.48 | 41.27 | 39.90 | **37.06** |
| | | MAPE (%) | 9.88 | 11.58 | 10.05 | 10.52 | 10.84 | 10.32 | 10.75 | 10.49 | 9.81 | 11.71 | 9.96 | **9.18** |

2019), **STID** (Shao et al., 2022a), **D²STGNN** (Shao et al., 2022c), **STAEF** (Liu et al., 2023a), **HimNet** (Dong et al., 2024), **STPGNN** (Kong et al., 2024), **PatchSTG** (Fang et al., 2025), and **BiST** (Ma et al., 2025a). For OOD generalization methods, we compare with **CauSTG** (Zhou et al., 2023), **STONE** (Wang et al., 2024), and **STOP** (Ma et al., 2025b). For pre-training methods, we include **STEP** (Shao et al., 2022b), **STD-MAE** (Gao et al., 2024), **FlashST** (Li et al., 2024), and **CrossST** (Liu & Zhang, 2025). For continual learning methods, we adopt **STKEC** (Wang et al., 2023a), **PECPM** (Wang et al., 2023b), **STRAP** (Zhang et al., 2025), and **EAC** (Chen & Liang, 2025).

**Implementation Details.** All experiments are conducted on a server equipped with an Intel Xeon Gold 5220 CPU and an NVIDIA Tesla V100 GPU (32GB). We use AdamW with a batch size of 64 and train up to 300 epochs with early stopping. We report mean absolute error (MAE), root mean squared error (RMSE), and mean absolute percentage error (MAPE) following prior work (Yu et al., 2018), and use MAE as the primary training objective. Unless otherwise stated, we set the stochastic environment perturbation ratio to $\rho = 0.2$ and the number of physical bases in the *Environment Basis Manifold* to $K = 4$.

Experimental details are in Section F. Code is available at GitHub.

*Table 2.* Few-shot cross-city transfer performance on CA-D3.

| Dataset | Metric | STEP | STD-MAE | FlashST | CrossST | STPDE |
|---|---|---|---|---|---|---|
| **CA-D3** | MAE | 18.48 | 17.74 | 18.49 | 17.51 | **16.25** |
| | RMSE | 29.78 | 29.11 | 29.49 | 27.84 | **26.54** |
| | MAPE | 14.84% | 14.27% | 14.78% | 14.79% | **13.39%** |

*Table 3.* Continual learning performance on PEMS-Stream.

| Dataset | Metric | STKEC | PECPM | STRAP | EAC | STPDE |
|---|---|---|---|---|---|---|
| **PEMS-Stream** | MAE | 16.96 | 16.86 | 16.88 | 15.67 | **11.40** |
| | RMSE | 27.56 | 27.37 | 27.35 | 25.30 | **18.95** |
| | MAPE | 21.50% | 21.73% | 22.17% | 20.42% | **16.01%** |

### 5.2. Performance Comparison

The results across the four evaluation settings are summarized in Tables 1–3. STPDE achieves the best *average* performance throughout, reducing the MAE (averaged over all datasets within each setting) by **1.5%**, **4.0%**, **7.2%**, and **27.2%** over the strongest runner-up, respectively. The gains are particularly pronounced in the more challenging regimes—OOD generalization, few-shot cross-city transfer, and continual learning—highlighting STPDE's robustness beyond standard in-distribution forecasting. These results substantiate our central claim: by formalizing spatio-temporal evolution as an environment-modulated PDE, the

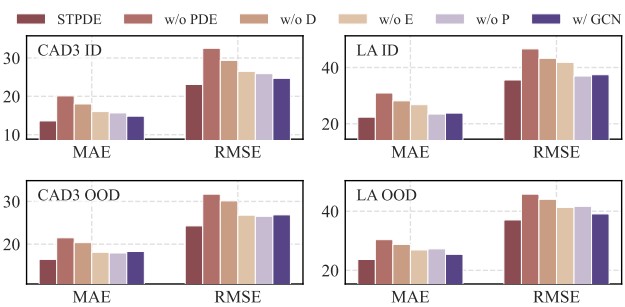

Figure 3. Results of ablation experiments.

Figure 4. Results of hyperparameter sensitivity analysis.

model more faithfully captures the underlying dynamics in complex spatio-temporal systems, leading to consistently lower forecasting error.

Interestingly, some methods tailored for spatio-temporal OOD generalization (e.g., STONE and STOP) can underperform even strong ID-oriented STGNNs (e.g., GWNet) in certain scenarios, consistent with recent benchmarking observations (Wang et al., 2025). We attribute this to two factors. First, some OOD objectives may yield "apparent" robustness primarily via *underfitting*: they reduce the relative degradation under shift but suffer larger absolute errors in both ID and OOD regimes. Second, several designs implicitly decouple temporal and spatial shifts, whereas real urban distribution changes are often strongly coupled; enforcing such a separation can mis-specify the learning problem and impair generalization.

For few-shot transfer and continual learning, STPDE differs fundamentally from approaches that rely on large-scale data to capture purely statistical modes. By leveraging a pre-trained *Invariant Diffusion Operator*, STPDE adapts to new domains mainly by reconstructing the *Environment Basis Manifold*, requiring only a small amount of target data for fine-tuning (see Section D for details). This behavior provides strong evidence that STPDE effectively disentangles environment-dependent factors from invariant propagation laws, enabling data-efficient adaptation under distribution shifts and evolving topologies.

### 5.3. Ablation Study

To validate the contribution of the physical components of STPDE and its core assumptions, we conduct systematic ablations on **CA-D3** and **LA** under both ID and OOD settings. We compare the full model against five variants: ❶ **w/o PDE** (replacing the PDE solver with an MLP), ❷ **w/o D** (removing the *Invariant Diffusion Operator*), ❸ **w/o E** (removing the *Environment Basis Manifold*), ❹ **w/o P** (removing the environment perturbation), and ❺ **w/ GCN** (using a GCN as the *Invariant Diffusion Operator*). Results are reported in Figure 3.

First, w/o PDE substantially degrades MAE on **CA-D3**, with a larger drop under OOD, indicating that explicitly modeling spatio-temporal dynamics as PDE evolution—rather than a purely regressive mapping—provides a crucial inductive bias for stability and transferability. Second, the consistent degradations of w/o D and w/o E support our "invariant dynamics–environment modulation" decoupling hypothesis: capturing generic evolution without conditioning on heterogeneous environments (w/o E) weakens cross-environment generalization, while modeling environmental differences without a stable global propagation backbone (w/o D) tends to induce memorization and fails to sustain performance under OOD. Moreover, w/o P exhibits the largest OOD performance drop, suggesting that environment perturbation is necessary to counter non-stationary distribution shifts and improve robustness. Finally, w/ GCN underperforms the *Invariant Diffusion Operator*, since GCNs primarily rely on local message passing and cannot realize the diffusion operator's global coupling/long-range propagation at comparable depth, which limits predictive accuracy. The ablation results on other datasets can be found in Section G.1.

### 5.4. Hyperparameter Study

To assess the stability of STPDE and the controllability of its key components, we conduct a sensitivity study over three hyperparameters: the hidden dimension $D \in \{64, 128, 192, 256, 320\}$, the stochastic environment perturbation ratio $\rho\% \in \{5\%, 10\%, 15\%, 20\%, 25\%\}$, and the number of physical bases maintained in the *Environment Basis Manifold* $K \in \{2, 4, 6, 8, 10\}$. We follow a one-factor-at-a-time protocol (varying one hyperparameter while keeping the others at their default values) and evaluate trends under both ID and OOD settings, with results presented in Figure 4 and Appendix Figure 9.

Overall, increasing $D$ improves accuracy but exhibits diminishing returns at $D = 320$ with higher computational cost; thus we recommend $D \leq 256$ for a favorable accuracy–efficiency trade-off. For perturbations, moderate stochasticity is essential for robustness, with the most stable OOD performance around $\rho\% \approx 20\%$: smaller $\rho\%$ provides insuf-

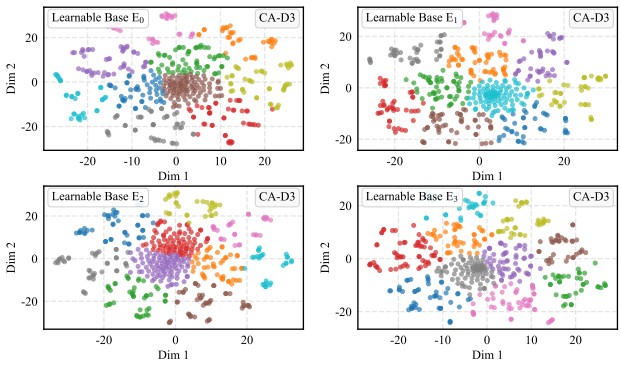

*Figure 5.* Visualizing the Environment Basis Manifold.

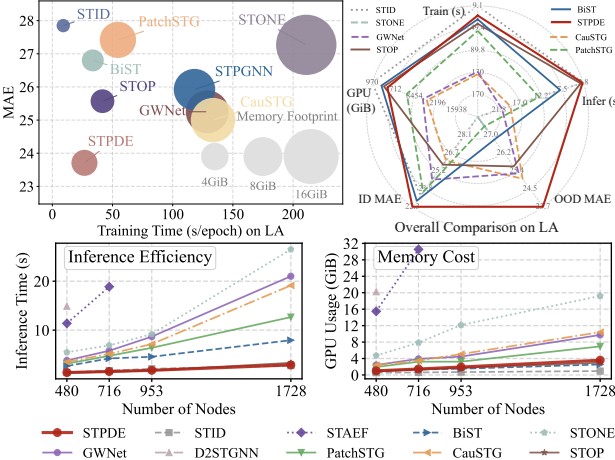

*Figure 6.* Computational efficiency comparison.

ficient regularization against non-stationary shifts, whereas larger $\rho\%$ (e.g., near $25\%$) injects excessive noise that interferes with learning nominal dynamics through the invariant operator, degrading both ID and OOD performance. Finally, increasing $K$ yields consistent gains initially but quickly plateaus, typically stabilizing around $K = 4$; further enlarging $K$ tends to introduce redundancy and may destabilize the gating selection mechanism, increasing training variance without consistent benefits.

## 5.5. Case Study

To improve the interpretability of the *Environment Basis Manifold* in STPDE, we visualize the learned bases on **CA-D3** using t-SNE under the default setting $K = 4$. As shown in Figure 5, the four learnable bases exhibit clearly different clustering patterns, suggesting that they are not redundant copies but capture distinct environment prototypes. Inspecting the routing weights further indicates that nodes with similar spatio-temporal dynamics tend to be assigned to the same (or nearby) bases, yielding an automatic grouping of node-wise patterns consistent with prior observations (Liu & Zhang, 2025; 2026). This behavior is enabled by two design choices: (i) stochastic perturbations together with a load-balancing loss encourage diverse and non-collapsed usage of bases, promoting specialization across heterogeneous environments; and (ii) a sparse Top-$k$ routing mechanism retrieves the most compatible bases and aggregates them with learned weights to modulate the invariant diffusion dynamics. Consequently, STPDE can adaptively compose environment-specific priors on top of a shared invariant backbone, which becomes particularly beneficial under distribution shifts. The visualization results on other datasets can be found in Section G.3.

## 5.6. Efficiency Study

To comprehensively evaluate practical efficiency, we benchmark STPDE against representative baselines on the largest dataset **LA** under a unified hardware setup and a fixed batch size of 32, reporting training time, inference time, and GPU memory footprint (see Figure 6). Models such as D²STGNN, STAEF, and GWNet, while competitive on certain tasks, incur substantially higher computation and memory costs due to expensive spatio-temporal operators, resulting in limited training and inference efficiency at scale. Moreover, as the number of nodes increases from **CA-D3** (480 nodes) to **LA** (1728 nodes), STPDE exhibits a markedly milder growth in cost; even compared with lightweight methods explicitly optimized for efficiency (e.g., PatchSTG and BiST), STPDE retains clear advantages in both training and inference time on **LA**, demonstrating strong scalability. Importantly, STPDE achieves these efficiency gains without sacrificing full-scenario capability: it maintains robust OOD generalization and few-shot adaptation, which are typically missing in purely lightweight designs, thereby offering a more balanced trade-off among accuracy, efficiency, and generalization.

## 6. Conclusion

In this paper, we present STPDE, a framework that reformulates spatio-temporal forecasting as the evolution of an inhomogeneous partial differential equation. By explicitly disentangling universal physical laws from environmental heterogeneity, STPDE employs an *Invariant Diffusion Operator* to capture global propagation dynamics with linear complexity. Complementarily, an *Environment Basis Manifold* with stochastic perturbations adaptively modulates these dynamics to accommodate heterogeneity and non-stationary shifts. Extensive experiments across diverse benchmarks confirm that STPDE not only achieves state-of-the-art performance on standard forecasting tasks but also demonstrates strong robustness in challenging OOD generalization, few-shot transfer, and continual learning scenarios, achieving a superior accuracy–efficiency trade-off.

## Impact Statement

This paper presents work whose goal is to advance the field of Machine Learning. There are many potential societal consequences of our work, none which we feel must be specifically highlighted here.

## Acknowledgment

This work was partly supported by the National Key Research and Development Program of China under Grant 2023YFB3002201, the National Natural Science Foundation of China under Grant 72342026, and Fundamental Research Funds for the Central Universities under Grant 2024-6-ZD-02.

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

# Appendix

## A. Notations

Table 4 summarizes the main symbols used in the paper and Appendix proofs.

*Table 4.* Core notations used in the main text and Appendix.

| Symbol | Meaning |
|---|---|
| $\Omega$ | Continuous spatial domain. |
| $\mathbf{x}, \mathbf{x}' \in \Omega$ | Target and source spatial locations. |
| $\tau$ | Diffusion time variable. |
| $\mathbf{u}(\mathbf{x}, \tau)$ | Diffusion field at location $\mathbf{x}$ and time $\tau$. |
| $\mathbf{u}_0(\mathbf{x})$ | Initial condition at $\tau = 0$. |
| $\nabla^2$ | Laplace operator on $\Omega$. |
| $\mathcal{K}(\mathbf{x}, \mathbf{x}'; \tau)$ | Continuous heat kernel (Green's function) propagating from $\mathbf{x}'$ to $\mathbf{x}$ over time $\tau$. |
| $\{\mathbf{x}_i\}_{i=1}^N$ | Discretized node locations. |
| $N$ | Number of nodes. |
| $\mathbf{u}_i(\tau)$ | Discretized state at node $i$. |
| $\mathbf{L} \in \mathbb{R}^{N \times N}$ | Discrete operator approximating $-\nabla^2$ (e.g., graph Laplacian). |
| $\mathbf{K}(\tau) \in \mathbb{R}^{N \times N}$ | Discrete heat kernel matrix. |
| $s_i(\tau)$ | Row-sum of $\mathbf{K}(\tau)$ at node $i$. |
| $\mathbf{D}_\mathbf{K}(\tau) \in \mathbb{R}^{N \times N}$ | Diagonal matrix formed from $\{s_i(\tau)\}_{i=1}^N$. |
| $\widetilde{\mathbf{K}}(\tau) \in \mathbb{R}^{N \times N}$ | Row-normalized kernel derived from $\mathbf{K}(\tau)$. |
| $\mathbf{h} \in \mathbb{R}^{N \times D}$ | Node feature matrix. |
| $D$ | Feature dimension used in query/key/value projections. |
| $\mathbf{h}_i \in \mathbb{R}^D$ | Node feature vector at node $i$. |
| $\mathbf{W}_q, \mathbf{W}_k, \mathbf{W}_v \in \mathbb{R}^{D \times D}$ | Projection matrices for query/key/value. |
| $\mathbf{h}_{q,i}, \mathbf{h}_{k,i}, \mathbf{h}_{v,i} \in \mathbb{R}^D$ | Query/key/value vectors at node $i$. |
| $\widehat{\mathbf{K}}(\tau) \in \mathbb{R}^{N \times N}$ | Feature-induced exponential kernel used to approximate $\mathbf{K}(\tau)$ up to row-wise normalization. |
| $\widehat{s}_i(\tau)$ | Row-sum of $\widehat{\mathbf{K}}(\tau)$ at node $i$. |
| $\mathbf{D}_{\widehat{\mathbf{K}}}(\tau) \in \mathbb{R}^{N \times N}$ | Diagonal matrix formed from $\{\widehat{s}_i(\tau)\}_{i=1}^N$. |
| $\alpha_{ij}$ | Softmax attention coefficient from Eq. (4). |
| $\mathbf{h}_i' \in \mathbb{R}^D$ | Updated representation at node $i$ after attention. |
| $\phi(\cdot)$ | Feature map in the separable kernel assumption for linear attention. |
| $\mathbf{G} \in \mathbb{R}^{D \times D}$ | Global context matrix used in linear attention. |

## B. Proof of Theorem 4.1

**Theorem 4.1.** *Let $\mathcal{G} = (\mathcal{V}, \mathbf{A})$ be a graph with $N$ nodes, and let $\mathcal{K}(\tau) \in \mathbb{R}^{N \times N}$ denote the discrete heat kernel at diffusion time $\tau$. Assume that $\mathcal{K}(\tau)$ can be approximated, up to row-wise normalization, by a feature-induced kernel $\widehat{\mathcal{K}}_{ij}(\tau) \propto \exp(\mathbf{h}_{q,i}^\top \mathbf{h}_{k,j})$. Then the Green's-function propagation $\mathbf{h}' = \mathcal{K}(\tau)\mathbf{h}_v$ is approximated by the global attention update in Eq. (4), where the row-normalized attention weights realize diffusion-type coupling over all nodes.*

*Proof.* We provide a detailed derivation that makes explicit the (i) Green's-function (heat-kernel) propagation view of diffusion, (ii) its discrete graph analogue, and (iii) the equivalence to global softmax attention under the stated kernel approximation.

**I. Continuous diffusion as Green's-function (heat-kernel) propagation.** Consider the diffusion PDE on a domain $\Omega$:

$$\frac{\partial \mathbf{u}(\mathbf{x}, \tau)}{\partial \tau} = \nabla^2 \mathbf{u}(\mathbf{x}, \tau), \tag{14}$$

with initial condition $\mathbf{u}(\mathbf{x}, 0) = \mathbf{u}_0(\mathbf{x})$. The classical analytic theory of parabolic PDEs gives the solution as a kernel

integral transform (the Green's-function (heat-kernel) representation):

$$\mathbf{u}(\mathbf{x}, \tau) = \int_{\Omega} \mathcal{K}(\mathbf{x}, \mathbf{x}'; \tau)\, \mathbf{u}_0(\mathbf{x}')\, d\mathbf{x}', \tag{15}$$

where $\mathcal{K}(\mathbf{x}, \mathbf{x}'; \tau)$ is the heat kernel (a Green's function) that propagates information from source location $\mathbf{x}'$ to target location $\mathbf{x}$ after time $\tau$. Equation (15) is the prototypical *global* coupling rule: every $\mathbf{x}'$ contributes to $\mathbf{x}$ through the kernel weight $\mathcal{K}(\mathbf{x}, \mathbf{x}'; \tau)$.

**II. Spatial discretization $\Rightarrow$ semi-discrete ODE system on a graph.** Let the spatial domain be discretized into $N$ nodes (locations) $\{\mathbf{x}_i\}_{i=1}^{N}$, and denote the discrete state by $\mathbf{u}_i(\tau) \approx \mathbf{u}(\mathbf{x}_i, \tau)$. Applying a standard *spatial* discretization of the Laplacian (e.g., finite differences, finite elements, or a graph-Laplacian discretization) yields a matrix operator $\mathbf{L}$ that approximates $-\nabla^2$ on the discretized domain. Keeping $\tau$ continuous (method of lines), the PDE in Eq. (14) reduces to the following *semi-discrete* ODE system:

$$\frac{d\mathbf{u}(\tau)}{d\tau} = -\mathbf{L}\,\mathbf{u}(\tau), \tag{16}$$

where $\mathbf{u}(\tau) \in \mathbb{R}^{N \times d}$ stacks node states (for $d$ channels). The solution is given by the (graph) heat kernel:

$$\mathbf{u}(\tau) = \exp(-\tau \mathbf{L})\, \mathbf{u}(0) =: \mathbf{K}(\tau)\, \mathbf{u}(0), \qquad \mathbf{K}(\tau) = \exp(-\tau \mathbf{L}) \in \mathbb{R}^{N \times N}, \tag{17}$$

i.e.,

$$\mathbf{u}_i(\tau) = \sum_{j=1}^{N} \mathbf{K}_{ij}(\tau)\, \mathbf{u}_j(0). \tag{18}$$

This is the discrete counterpart of the continuous Green's-function integral in Eq. (15).

**III. Row-wise normalization written explicitly (no shorthand).** In many diffusion and propagation constructions, it is convenient to work with a row-stochastic (row-normalized) kernel to interpret weights as normalized influence coefficients. Define the row-sum and the corresponding diagonal normalizer:

$$s_i(\tau) \triangleq \sum_{z=1}^{N} \mathbf{K}_{iz}(\tau), \qquad \mathbf{D}_{\mathbf{K}}(\tau) \triangleq \mathrm{diag}\big(s_1(\tau), \dots, s_N(\tau)\big). \tag{19}$$

Define the row-normalized kernel by explicit matrix normalization:

$$\widetilde{\mathbf{K}}(\tau) \triangleq \mathbf{D}_{\mathbf{K}}(\tau)^{-1}\mathbf{K}(\tau), \qquad \widetilde{\mathbf{K}}_{ij}(\tau) = \frac{\mathbf{K}_{ij}(\tau)}{\sum_{z=1}^{N} \mathbf{K}_{iz}(\tau)}. \tag{20}$$

Then $\sum_{j=1}^{N} \widetilde{\mathbf{K}}_{ij}(\tau) = 1$ for each $i$, and propagation reads:

$$\mathbf{u}_i(\tau) = \sum_{j=1}^{N} \widetilde{\mathbf{K}}_{ij}(\tau)\, \mathbf{u}_j(0). \tag{21}$$

**IV. Feature-induced approximation $\Rightarrow$ softmax attention weights (matching the main text).** Now map the observed field to latent node states $\mathbf{h} \in \mathbb{R}^{N \times D}$. Define $(\mathbf{h}_{q,i}, \mathbf{h}_{k,i}, \mathbf{h}_{v,i})$ exactly as in Eq. (3). (We do *not* redefine them here to avoid notation mismatch with the main text.)

Assume, as stated in the theorem, that the discrete heat kernel is approximated *up to row-wise normalization* by a feature-induced exponential kernel consistent with Eq. (4):

$$\widehat{\mathbf{K}}_{ij}(\tau) \propto \exp\!\left(\frac{\mathbf{h}_{q,i}^{\top}\mathbf{h}_{k,j}}{\sqrt{D}}\right). \tag{22}$$

Define the unnormalized kernel matrix and its row-sums explicitly:

$$\widehat{\mathbf{K}}_{ij}(\tau) \triangleq \exp\!\left(\frac{\mathbf{h}_{q,i}^{\top}\mathbf{h}_{k,j}}{\sqrt{D}}\right), \qquad \widehat{s}_i(\tau) \triangleq \sum_{z=1}^{N} \widehat{\mathbf{K}}_{iz}(\tau), \qquad \mathbf{D}_{\widehat{\mathbf{K}}}(\tau) \triangleq \mathrm{diag}\big(\widehat{s}_1(\tau), \dots, \widehat{s}_N(\tau)\big). \tag{23}$$

Performing the corresponding row-wise normalization (cf. Eq. (20)) yields the attention weights:

$$\alpha_{ij} \triangleq \frac{\widehat{\mathbf{K}}_{ij}(\tau)}{\sum_{z=1}^{N} \widehat{\mathbf{K}}_{iz}(\tau)} = \frac{\exp\left(\frac{\mathbf{h}_{q,i}^{\top}\mathbf{h}_{k,j}}{\sqrt{D}}\right)}{\sum_{z=1}^{N} \exp\left(\frac{\mathbf{h}_{q,i}^{\top}\mathbf{h}_{k,z}}{\sqrt{D}}\right)}. \tag{24}$$

By construction, $\sum_j \alpha_{ij} = 1$ and $\alpha_{ij} > 0$ for all $i, j$. Thus, replacing $\widetilde{\mathbf{K}}_{ij}(\tau)$ in Eq. (21) with $\alpha_{ij}$ and using $\mathbf{h}_{v,j}$ as the propagated quantity, we obtain:

$$\mathbf{h}_i' = \sum_{j=1}^{N} \alpha_{ij}\,\mathbf{h}_{v,j} = \sum_{j=1}^{N} \frac{\exp\left(\frac{\mathbf{h}_{q,i}^{\top}\mathbf{h}_{k,j}}{\sqrt{D}}\right)}{\sum_{z=1}^{N} \exp\left(\frac{\mathbf{h}_{q,i}^{\top}\mathbf{h}_{k,z}}{\sqrt{D}}\right)}\,\mathbf{h}_{v,j}, \tag{25}$$

which is exactly the global softmax attention form in Eq. (4).

**V. "Up to row-wise normalization" made explicit at the matrix level.** The theorem hypothesis "approximated up to row-wise normalization" can be written explicitly as:

$$\mathbf{D}_{\widehat{\mathbf{K}}}(\tau)^{-1}\widehat{\mathbf{K}}(\tau) \approx \mathbf{D}_{\mathbf{K}}(\tau)^{-1}\mathbf{K}(\tau) = \widetilde{\mathbf{K}}(\tau), \tag{26}$$

where $\mathbf{D}_{\mathbf{K}}(\tau)$ is defined in Eq. (19) and $\mathbf{D}_{\widehat{\mathbf{K}}}(\tau)$ is defined in Eq. (23). Applying both sides to the same node signal (here, the value matrix $\mathbf{H}_v$) yields an attention-style approximation to row-normalized heat-kernel propagation.

**VI. "Global diffusion coupling" interpretation.** Because $\alpha_{ij} > 0$ for all $j$ (for fixed $i$), every node $j$ contributes to the update of node $i$; this implements *global* coupling, analogous to Eq. (15). Hence the Green's-function propagation induced by $\mathcal{K}(\cdot, \cdot; \tau)$ admits an attention-style approximation, completing the proof. $\square$

## C. Proof of Theorem 4.2

**Theorem 4.2.** *Assume the kernel admits the separable form in Eq. (5), and define $\mathbf{G}$ as in Eq. (6). Then the global kernel propagation $\mathbf{h}_i' = \sum_{j=1}^{N} \phi_q(\mathbf{h}_{q,i})^{\top}\phi_k(\mathbf{h}_{k,j})\,\mathbf{h}_{v,j}$ can be computed exactly by the two-step procedure in Eqs. (6)–(7), thereby reducing the attention-style global coupling from $\mathcal{O}(N^2)$ to $\mathcal{O}(ND^2)$ time.*

*Proof.* We show that when the kernel is separable, global propagation can be computed exactly via associativity, without materializing the $N \times N$ affinity matrix.

**I. Start from separable kernel propagation.** Assume the kernel admits the separable form in Eq. (5):

$$\widehat{\mathbf{K}}_{ij} = \phi_q(\mathbf{h}_{q,i})^{\top}\phi_k(\mathbf{h}_{k,j}), \tag{27}$$

where $\phi(\cdot)$ is a feature map. Consider the global kernel propagation (attention-style but without the explicit $N \times N$ matrix):

$$\mathbf{h}_i' = \sum_{j=1}^{N} \left(\phi_q(\mathbf{h}_{q,i})^{\top}\phi_k(\mathbf{h}_{k,j})\right)\mathbf{h}_{v,j}. \tag{28}$$

**II. Use associativity to factor out the $i$-dependent term.** The scalar coefficient $\phi_q(\mathbf{h}_{q,i})^{\top}\phi_k(\mathbf{h}_{k,j})$ can be rewritten by swapping the inner product:

$$\phi_q(\mathbf{h}_{q,i})^{\top}\phi_k(\mathbf{h}_{k,j}) = \phi_k(\mathbf{h}_{k,j})^{\top}\phi_q(\mathbf{h}_{q,i}). \tag{29}$$

Substitute Eq. (29) into Eq. (28):

$$\mathbf{h}_i' = \sum_{j=1}^{N} \left(\phi_k(\mathbf{h}_{k,j})^{\top}\phi_q(\mathbf{h}_{q,i})\right)\mathbf{h}_{v,j}. \tag{30}$$

Now observe that $\left(\phi_k(\mathbf{h}_{k,j})^{\top}\phi_q(\mathbf{h}_{q,i})\right)\mathbf{h}_{v,j}$ is a scalar multiplying a vector, which can be expressed via an outer product:

$$\left(\phi_k(\mathbf{h}_{k,j})^{\top}\phi_q(\mathbf{h}_{q,i})\right)\mathbf{h}_{v,j} = \left(\mathbf{h}_{v,j}\,\phi_k(\mathbf{h}_{k,j})^{\top}\right)\phi_q(\mathbf{h}_{q,i}). \tag{31}$$

(Here $\mathbf{h}_{v,j} \in \mathbb{R}^D$ is treated as a column vector, so $\mathbf{h}_{v,j}\,\phi_k(\mathbf{h}_{k,j})^\top \in \mathbb{R}^{D \times D}$.) Plugging Eq. (31) into Eq. (30) yields:

$$\mathbf{h}'_i = (\sum_{j=1}^N \mathbf{h}_{v,j}\,\phi_k(\mathbf{h}_{k,j})^\top)\phi_q(\mathbf{h}_{q,i}). \tag{32}$$

**III. Match the paper's "aggregate–distribute" form.** Define the global context matrix exactly as in Eq. (6) (note that this is consistent with the transpose convention used in Eq. (7)):

$$\mathbf{G} := \sum_{j=1}^N \phi_k(\mathbf{h}_{k,j})\,\mathbf{h}_{v,j}^\top \in \mathbb{R}^{D \times D}. \tag{33}$$

Taking transpose gives:

$$\mathbf{G}^\top = \sum_{j=1}^N \mathbf{h}_{v,j}\,\phi_k(\mathbf{h}_{k,j})^\top. \tag{34}$$

Substitute Eq. (34) into Eq. (32):

$$\mathbf{h}'_i = \mathbf{G}^\top \phi_q(\mathbf{h}_{q,i}), \tag{35}$$

which is exactly the two-step procedure in Eqs. (6)–(7). Therefore, the computation is *algebraically identical* to the original global propagation in Eq. (28).

**IV. Complexity.** Computing $\mathbf{G}$ in Eq. (33) requires summing $N$ outer products $\phi_k(\mathbf{h}_{k,j})\,\mathbf{h}_{v,j}^\top \in \mathbb{R}^{D \times D}$. Each outer product costs $\mathcal{O}(D^2)$, so forming $\mathbf{G}$ costs $\mathcal{O}(ND^2)$ time and $\mathcal{O}(D^2)$ memory. Then, retrieving $\mathbf{h}'_i = \mathbf{G}^\top \phi_q(\mathbf{h}_{q,i})$ costs $\mathcal{O}(D^2)$ per node, hence $\mathcal{O}(ND^2)$ for all nodes. Crucially, this avoids constructing the $N \times N$ affinity matrix (which would cost $\mathcal{O}(N^2)$ time and memory).

**Conclusion.** Under the separable kernel assumption, the attention-style global coupling can be computed exactly by associativity via the aggregate–distribute decomposition, reducing the quadratic dependence on $N$ to linear (for fixed $D$). □

# D. Algorithm Workflow

This section summarizes the end-to-end workflow of STPDE under different learning scenarios, corresponding to Algorithms 1–3 in the appendix. Across all settings, STPDE follows the encode–solve–decode pipeline and inference routines described in the main paper, and outputs predictions in the same format as standard spatio-temporal forecasting models. The key distinction among tasks lies in whether (and how) we conduct pre-training, fine-tuning, and incremental adaptation.

### D.1. ID and OOD Forecasting

For in-distribution (ID) and out-of-distribution (OOD) forecasting tasks, the workflow is given in Algorithm 1. Since these settings do not require additional training stages beyond the standard model usage, STPDE directly performs forward inference according to the canonical pipeline in the main paper. Concretely, given historical spatio-temporal observations, STPDE constructs the required pattern representations and produces multi-step forecasts through a single forward pass. No extra pre-training or task-specific fine-tuning is introduced in Algorithm 1; thus, the procedure aligns with the regular inference protocol used throughout our experiments.

### D.2. Cross-City Few-Shot Forecasting

For cross-city few-shot forecasting, the workflow is described in Algorithm 2. The goal is to transfer generalizable knowledge from a source domain (with abundant data and broader pattern diversity) to a target domain where only limited labeled samples are available.

**Stage I: Source-domain pre-training.** STPDE is first pre-trained on a large-scale source-domain dataset. This stage aims to extract universal spatio-temporal patterns from data with richer variability and larger coverage, yielding transferable representations that can serve as a robust initialization for downstream adaptation.

---

**Algorithm 1** Training of `STPDE` via Physics-Constrained Evolution

---

**Require:** Training set $(\mathcal{X}, \mathcal{Y})$; Top-$k$ sparsity $k_s$; perturbation ratio $\rho$; weight $\lambda$; lr $\eta$

1: **Trainable:** $\Theta = \{\phi_{\text{in}}, \phi_{\text{out}}, \mathbf{W}_q, \mathbf{W}_k, \mathbf{W}_v, \Phi = \{\mathbf{E}_k\}_{k=1}^K, \mathcal{R}, \text{AdaLN}, \text{FFN}, \mathbf{e}_{\text{avg}}\}$; invariant operator params $\theta_{\text{inv}}$

2: **Helper:** `TopKSoftmax`$(\cdot, k_s)$ returns sparse row-wise weights; `PerturbBases`$(\Phi, \rho)$ returns $\{\tilde{\mathbf{E}}_k^{(i)}\}$

3: **while** not converged **do**

4:     **for** mini-batch $(\mathbf{x}, \mathbf{y})$ from $(\mathcal{X}, \mathcal{Y})$ **do**

5:         // Stage 1: Lift to latent field

6:         $\mathbf{H} \leftarrow \phi_{\text{in}}(\mathbf{x})$ $\{\mathbf{H} \in \mathbb{R}^{N \times D}\}$

7:         // Stage 2: Stochastic environment perturbation (Section 4.3)

8:         $\{\tilde{\mathbf{E}}_k^{(i)}\}_{k=1}^K \leftarrow$ `PerturbBases`$(\Phi, \rho)$ $\{$Choose from $\mathbf{E}_k$, permuted $\Pi(\mathbf{E}_k)$, or $\mathbf{e}_{\text{avg}}\}$

9:         // Stage 3: Environment inference via sparse routing (Eq. 10–11)

10:        $\mathbf{Z} \leftarrow \mathcal{R}(\mathbf{H})$ $\{\mathbf{Z} \in \mathbb{R}^{N \times K}$ (logits)$\}$

11:        $\mathbf{W} \leftarrow$ `TopKSoftmax`$(\mathbf{Z}, k_s)$ $\{\mathbf{W} = [w_{i,k}] \in \mathbb{R}^{N \times K}\}$

12:        $\mathbf{E}[i,:] \leftarrow \sum_{k=1}^K w_{i,k} \tilde{\mathbf{E}}_k^{(i)}[i,:]$ $\forall i$ $\{\mathbf{E} \in \mathbb{R}^{N \times D}$ is node-wise environment embedding$\}$

13:        // Stage 4: Invariant diffusion operator (linear attention form)

14:        $\mathbf{Q}, \mathbf{K}, \mathbf{V} \leftarrow \mathbf{H}\mathbf{W}_q, \mathbf{H}\mathbf{W}_k, \mathbf{H}\mathbf{W}_v$

15:        $\mathbf{G} \leftarrow \sum_{j=1}^N \phi_k(\mathbf{K}_j) \mathbf{V}_j^\top$ $\{\mathbf{G} \in \mathbb{R}^{D \times D}\}$

16:        $\mathbf{H}_i' \leftarrow \mathbf{G}^\top \phi_q(\mathbf{Q}_i)$ $\forall i$ $\{\mathbf{H}' \in \mathbb{R}^{N \times D}\}$

17:        // Stage 5: Environment modulation + residual evolution (Eq. 12)

18:        $\boldsymbol{\gamma}, \boldsymbol{\beta} \leftarrow$ AdaLN$(\mathbf{E})$ $\{\gamma, \beta \in \mathbb{R}^{N \times D}\}$

19:        $\tilde{\mathbf{H}} \leftarrow \mathbf{H} + \boldsymbol{\gamma} \odot \mathbf{H}' + \boldsymbol{\beta} + \text{FFN}(\mathbf{H}')$

20:        // Stage 6: Decode + losses

21:        $\hat{\mathbf{y}} \leftarrow \phi_{\text{out}}(\tilde{\mathbf{H}})$

22:        $\mathcal{L}_{\text{task}} \leftarrow$ MAE$(\hat{\mathbf{y}}, \mathbf{y})$

23:        $P_k \leftarrow \frac{1}{|B|N} \sum_{x \in B} \sum_{i=1}^N w_{i,k}$ $\{$marginal usage$\}$

24:        $\mathcal{L}_{\text{aux}} \leftarrow K \sum_{k=1}^K P_k^2 - 1$ $\{$Load balancing$\}$

25:        $\Theta \leftarrow \Theta - \eta \nabla_\Theta(\mathcal{L}_{\text{task}} + \lambda \mathcal{L}_{\text{aux}})$

26:     **end for**

27: **end while**

---

**Stage II: Target-domain adaptation via Environment Basis Manifold reconstruction.** On the target downstream dataset, we freeze all branches of `STPDE` *except* the *Environment Basis Manifold*. We then reconstruct (i.e., re-parameterize and optimize) the Environment Basis Manifold to match the target environment, enabling explicit environment adaptation while keeping the rest of the model stable. By fine-tuning only this lightweight, environment-specific component, `STPDE` adapts rapidly to the target city and achieves strong performance in few-shot regimes. This design provides an effective balance between adaptation capacity and overfitting resistance under limited data.

### D.3. Continual Spatio-Temporal Learning

For continual learning, the workflow is given in Algorithm 3. In real-world deployments, spatio-temporal data often arrive incrementally: as time progresses, traffic networks expand due to new sensor deployments, which induces an evolving graph topology and a continuously growing node set. This incremental setting introduces two major challenges: catastrophic forgetting and distribution shift across increments.

**Initial increment: standard training.** At the first incremental period, `STPDE` is trained in the standard supervised manner on the available graph and data stream, establishing a strong base model.

**Subsequent increments: freeze-and-expand adaptation.** For each later incremental period, we freeze all model branches except the Environment Basis Manifold, and adapt to the expanded topology by *expanding the trainable parameters* of the Environment Basis Manifold. This expansion equips `STPDE` with additional capacity to encode newly introduced nodes and structural changes, while preserving previously learned knowledge stored in the frozen branches. As a result, `STPDE` substantially mitigates catastrophic forgetting and remains robust to incremental distribution shifts, enabling efficient and stable learning under continuously evolving spatio-temporal graphs.

---

**Algorithm 2** Few-Shot City Transfer via Environment Basis Manifold Reconstruction

---

**Require:** Pre-trained backbone $\Theta^* = \{\theta_{\text{inv}}^*, \Phi^*, \dots\}$, target data $\mathcal{D}_{\text{tgt}}$ (few samples)
**Ensure:** Fine-tuned forecast $\mathbf{y}_{\text{tgt}}$
1: // Step 1: Decoupling Physics and Environment
2: Freeze invariant backbone: $\theta_{\text{inv}} \leftarrow \text{freeze}(\Theta^*)$ {Universal laws are shared}
3: Reconstruct target manifold: $\Phi', \mathcal{R}' \leftarrow \text{Init}(\text{TargetNodes})$ {Re-initialize environment basis}
4: // Step 2: Manifold Alignment (Fine-tuning)
5: **while** not converged **do**
6:     **for** batch $(\mathbf{x}, \mathbf{y}) \in \mathcal{D}_{\text{tgt}}$ **do**
7:         $\mathbf{h} \leftarrow \phi_{\text{in}}(\mathbf{x})$
8:         /* Infer environment from new manifold */
9:         $w \leftarrow \text{Softmax}(\text{Top-}k(\mathcal{R}'(\mathbf{h})))$ {Get sparse routing weights}
10:         $\mathbf{e} \leftarrow \sum_k w_k \Phi_k'$ {RouteAndAgg}
11:         /* Forward with frozen physics */
12:         $\hat{\mathbf{y}} \leftarrow \texttt{STPDE\_Forward}(\mathbf{x}; \theta_{\text{inv}}, \mathbf{e})$
13:         /* Optimization with Load Balancing */
14:         $\mathcal{L}_{\text{task}} \leftarrow \text{MAE}(\hat{\mathbf{y}}, \mathbf{y})$
15:         $\mathcal{L}_{\text{aux}} \leftarrow \sum_k (P_k - 1/K)^2$ {Prevent basis collapse}
16:         Update $\Phi', \mathcal{R}'$ via $\nabla(\mathcal{L}_{\text{task}} + \lambda \mathcal{L}_{\text{aux}})$
17:     **end for**
18: **end while**

---

# E. Extended Related Work

## E.1. Spatio-Temporal Forecasting

Spatio-temporal (ST) forecasting is a core problem in intelligent transportation systems (ITS), where the goal is to predict future traffic states by jointly modeling temporal dynamics and spatial interactions among sensors or regions. Spatio-temporal graph neural networks (STGNNs) have become a dominant paradigm due to their ability to capture complex, non-Euclidean spatial dependencies together with multi-scale temporal patterns. Early STGNNs typically coupled graph convolutions with sequence models. Representative works such as DCRNN (Li et al., 2018) and STGCN (Yu et al., 2018) established canonical designs: the former integrates diffusion-based graph propagation with recurrent units, while the latter proposes an entirely convolutional framework that combines graph convolution for spatial correlation and temporal convolutions to replace recurrent computation, improving training efficiency.

A notable line of research relaxes the reliance on a fixed, predefined topology by learning latent spatial structures from data. Graph WaveNet (GWNet) (Wu et al., 2019) introduces an adaptive adjacency matrix to capture hidden spatial dependencies, and employs dilated causal convolutions to model long-range temporal dependencies, substantially improving robustness when the true graph is unknown or partially observed. Beyond topology learning, recent methods emphasize representation quality and disentanglement. D$^2$STGNN (Shao et al., 2022c) advocates a decoupled spatio-temporal formulation that separates traffic signals into diffusion and intrinsic components and leverages dynamic graph learning to capture time-varying spatial dependencies.

Interestingly, several works question whether architectural complexity is always necessary. STID (Shao et al., 2022a) demonstrates that a carefully designed MLP baseline can achieve competitive performance by introducing spatial and temporal identity embeddings to address the indistinguishability of samples across nodes and time steps. Building on this insight, STAEformer (Liu et al., 2023a) further highlights the importance of adaptive spatio-temporal embeddings, showing that an otherwise vanilla Transformer can reach state-of-the-art accuracy while maintaining architectural simplicity.

Recent advances additionally focus on heterogeneity, key-node modeling, scalability, and alternative learning paradigms (Chen et al., 2026). HimNet (Dong et al., 2024) proposes heterogeneity-informed meta-parameter learning, clustering context via learned spatio-temporal embeddings and dynamically instantiating model parameters from a meta-parameter pool to explicitly exploit heterogeneity. From a topological perspective, STPGNN (Kong et al., 2024) argues that pivotal (hub-like) nodes exhibit unique dependency structures and introduces dedicated modules for pivotal node identification and specialized graph convolution. To address computational bottlenecks in large-scale networks,

---

**Algorithm 3** Continual Learning via Environment Basis Manifold Expansion

---

**Require:** Initial model $\Theta^{(1)}$, data streams $\{\mathcal{D}^{(m)}\}_{m=1}^{M}$
**Ensure:** Optimized model parameters for all tasks
 1: // Phase 1: Initial Training (Base Task)
 2: $\Theta^{(1)*} \leftarrow \arg\min_{\Theta}(\mathcal{L}_{\text{task}} + \lambda\mathcal{L}_{\text{aux}})(\mathcal{D}^{(1)})$
 3: // Phase 2: Incremental Learning (New Tasks)
 4: **for** $m = 2$ to $M$ **do**
 5:     Freeze backbone: $\theta_{\text{inv}} \leftarrow \theta_{\text{inv}}^{(1)*}$ {Lock physical laws}
 6:     /* Expand Manifold for new nodes $\Delta\mathcal{V}$ */
 7:     Expand Basis: $\Phi^{(m)} \leftarrow \Phi^{(m-1)} \parallel \Delta\Phi^{(m)}$
 8:     Expand Router: $\mathcal{R}^{(m)} \leftarrow \mathcal{R}^{(m-1)} \parallel \Delta\mathcal{R}^{(m)}$
 9:     Initialize $\Delta\Phi^{(m)}, \Delta\mathcal{R}^{(m)}$ for $\Delta\mathcal{V}$
10:     **while** not converged **do**
11:       **for** batch $(\mathbf{x}, \mathbf{y}) \in \mathcal{D}^{(m)}$ **do**
12:         $\mathbf{h} \leftarrow \phi_{\text{in}}(\mathbf{x})$
13:         $w \leftarrow \text{Softmax}(\text{Top-}k(\mathcal{R}^{(m)}(\mathbf{h})))$
14:         $\mathbf{e} \leftarrow \sum_k w_k \Phi_k^{(m)}$
15:         $\hat{\mathbf{y}} \leftarrow \texttt{STPDE\_Forward}(\mathbf{x}; \theta_{\text{inv}}, \mathbf{e})$
16:         /* Ensure balanced usage of expanded basis */
17:         $\mathcal{L}_{\text{task}} \leftarrow \text{MAE}(\hat{\mathbf{y}}, \mathbf{y})$
18:         $\mathcal{L}_{\text{aux}} \leftarrow \sum_k (P_k - 1/K^{(m)})^2$
19:         Update $\Phi^{(m)}, \mathcal{R}^{(m)}$ via $\nabla(\mathcal{L}_{\text{task}} + \lambda\mathcal{L}_{\text{aux}})$
20:       **end for**
21:     **end while**
22: **end for**

---

PatchSTG (Fang et al., 2025) adopts irregular spatial patching (e.g., via leaf KD-trees) to partition unevenly distributed sensors into balanced patches, and combines local–global attention to achieve substantial gains in speed and memory efficiency. Finally, BiST (Ma et al., 2025a) challenges the conventional one-way forecasting pipeline by introducing a lightweight bidirectional formulation with a forward spatio-temporal learner and a backward residual correction stage, effectively mitigating spatio-temporal bias between inputs and targets at low cost.

### E.2. Spatio-Temporal OOD Generalization

While ST forecasting models can achieve impressive in-distribution (ID) performance, real-world deployment often faces distribution shifts across time, space, and exogenous conditions. This spatio-temporal out-of-distribution (ST-OOD) setting reveals a persistent gap between benchmark accuracy and practical robustness, where strong ID models may degrade sharply under cross-year, cross-region, or structural shifts. Recent studies have begun to systematize this challenge by constructing benchmarks and analyzing failure modes, often observing that even simple MLP-style baselines can be surprisingly competitive under severe OOD settings (Wang et al., 2025).

A growing body of work leverages causality and invariance principles to improve generalization. CauSTG (Zhou et al., 2023) treats different time steps as environments and aims to extract spatio-temporal relations that remain invariant across environments. From the perspective of structural causal models, CaST (Xia et al., 2023) separates invariant components from time-varying environments via back-door adjustment, and further models dynamic spatial causality using front-door adjustment. To jointly address spatial and temporal shifts, STONE (Wang et al., 2024) proposes a gated Transformer-style architecture designed to learn robust node dependencies that are less sensitive to environmental changes.

Complementary to causal learning, another line of research focuses on robust architectures and explicit modeling of confounders. STEVE (Ji et al., 2025) targets continuous or unspecified external confounders (e.g., extreme weather) with a self-supervised framework that constructs a basis confounder bank and represents complex influences through linear composition. At the architectural level, STOP (Ma et al., 2025b) argues that heavy GNN stacks are not strictly necessary for OOD robustness and proposes an efficient spatio-temporal MLP processor enhanced with centralized messaging and graph perturbation, significantly strengthening resilience to spatio-temporal disturbances.

### E.3. Spatio-Temporal Pre-Training

Inspired by the success of large-scale pre-training in NLP and vision, spatio-temporal pre-training has emerged as a promising approach to improve representation learning, long-range dependency modeling, and transferability across domains (Liu et al., 2025c;a). A prominent early direction adapts masked autoencoding (MAE) to time series and spatio-temporal data. STEP (Shao et al., 2022b) introduces a Transformer-based pre-training model (TSFormer) with masked reconstruction objectives to learn segment-level representations from long histories, benefiting downstream STGNN forecasting by strengthening long-term temporal modeling.

Subsequent works enrich the pre-training objective and structure to better capture complex spatio-temporal interactions. GPT-ST (Li et al., 2023) explores generative pre-training with hierarchical (hyper-)graph modeling and adaptive masking, enabling a curriculum-like learning process that captures both intra-cluster and inter-cluster dependencies. To address heterogeneity-induced "spatio-temporal mirage" effects, STD-MAE (Gao et al., 2024) proposes a decoupled MAE strategy, separately masking and reconstructing along spatial and temporal axes to enhance interpretability and robustness under heterogeneous patterns.

More recent research emphasizes efficient and effective transfer to diverse downstream tasks. FlashST (Li et al., 2024) introduces prompt learning into spatio-temporal forecasting, combining contextual distillation and unified distribution mapping to mitigate the mismatch between pre-training and fine-tuning distributions and to enable fast adaptation across cities. CrossST (Liu & Zhang, 2025) proposes a pattern-bank-based framework with pattern disentanglement, separating universal cross-region patterns from region-specific ones, thereby improving cross-city transfer while reducing computation.

### E.4. Continual Spatio-Temporal Learning

Real-world spatio-temporal data streams evolve continuously, and traffic networks may expand as new sensors or regions are added. Continual spatio-temporal learning aims to adapt to new patterns while retaining previously acquired knowledge, addressing catastrophic forgetting under non-stationary dynamics (Miao et al., 2025).

Early attempts borrow general continual learning techniques. TrafficStream (Chen et al., 2021) introduces continual learning for streaming traffic forecasting by detecting distribution changes (e.g., via divergence measures) and consolidating knowledge through experience replay and regularization. However, replay may be costly or infeasible due to privacy and storage constraints, motivating replay-free strategies. PECPM (Wang et al., 2023b) maintains a representative pattern bank and performs pattern expansion and consolidation to accommodate new or conflicting nodes while mitigating forgetting without direct access to raw historical data. Similarly, STKEC (Wang et al., 2023a) proposes influence-based knowledge expansion to identify critical nodes for efficient adaptation, and employs memory-augmented modules to explicitly preserve long- and short-term spatio-temporal patterns.

To further improve efficiency as networks grow, recent works explore parameter-efficient adaptation paradigms. EAC (Chen & Liang, 2025) adopts a prompt-based continual forecasting framework that freezes the backbone and maintains a continuous prompt pool, following an "expand-and-compress" principle to capture heterogeneity while preventing parameter explosion via low-rank compression. Under ST-OOD continual settings, STRAP (Zhang et al., 2025) proposes retrieval-augmented pattern learning with a multi-dimensional key–value library (spatial, temporal, and interaction patterns), retrieving and adaptively fusing historical patterns most similar to current inputs to alleviate forgetting and strengthen robustness.

## F. Experimental Details

### F.1. Dataset Details

**Datasets.** We evaluate STPDE on two mainstream spatio-temporal traffic forecasting benchmarks with complementary characteristics: **LargeST** (Liu et al., 2023b) and **PEMS-Stream** (Chen et al., 2021). Their key statistics are summarized in Table 5 and Table 6.

**LargeST (California Traffic Benchmark).** LargeST consists of multi-year traffic sensor measurements collected from California's Performance Measurement System (PeMS). Following the standard LargeST setup, we consider four representative sub-regions with diverse topology scales and spatial heterogeneity: ❶ **CA-D3** (North Central, $N=480$), a medium-sized highway-dominant network; ❷ **SD** (San Diego, $N=716$), a large metropolitan network; ❸ **Orange** (Orange County, $N=953$), a dense urban grid; ❹ **LA** (Greater Los Angeles, $N=1,728$), an extra-large and highly heterogeneous network. All raw sensor readings are aggregated into 5-minute intervals (288 steps per day). Missing values are imputed via

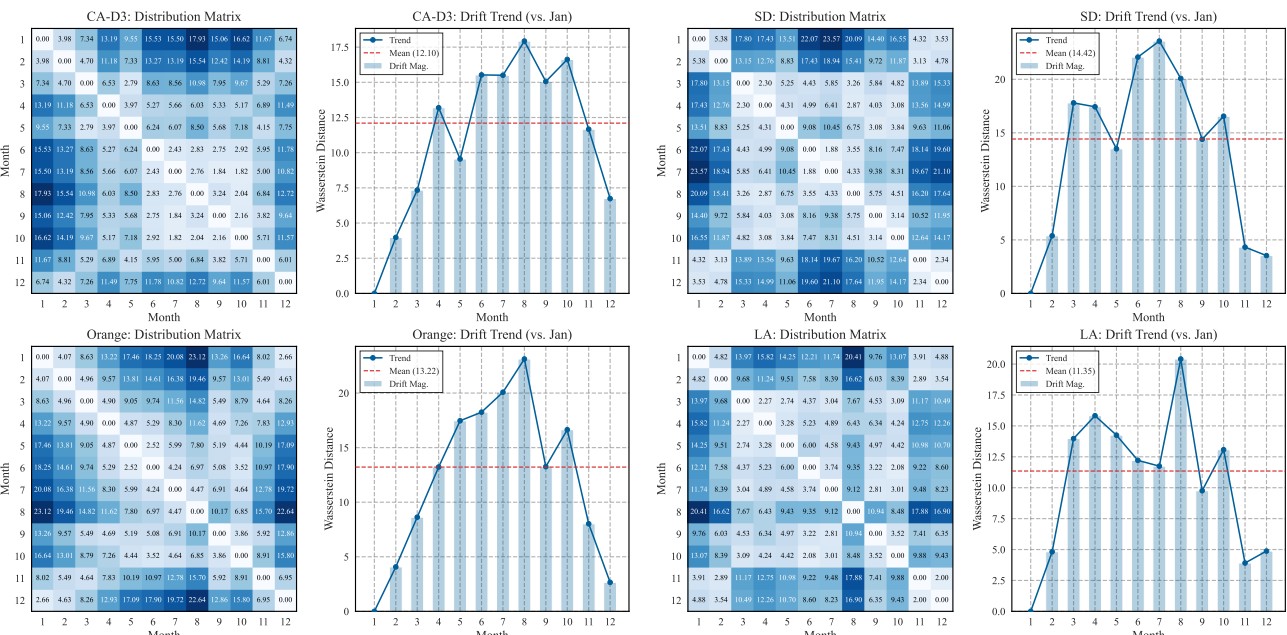

*Figure 7.* Visualization of month-to-month distribution shifts using Wasserstein distance matrices and drift trends across SD, CA-D3, Orange, and LA datasets.

*Table 5.* Overview of continual spatio-temporal forecasting datasets.

| Dataset | Domain | Time Range | Period | Node Expansion | Frequency |
|---------|--------|-----------|--------|----------------|-----------|
| **PEMS-Stream** | Traffic | 07/10/2011 - 09/08/2017 | 7 | $655 \rightarrow 715 \rightarrow 786$ $\rightarrow 822 \rightarrow 834 \rightarrow 850$ $\rightarrow 871$ | 5 min |

linear interpolation. We apply Z-score normalization to each channel using training-set statistics: $x' = (x - \mu)/\sigma$, where $\mu$ and $\sigma$ are computed on the training split only.

**Graph construction.** For methods requiring an explicit road-network prior, we construct a weighted adjacency matrix $\mathbf{A}$ based on real geodesic distances between sensors. Specifically, we use a thresholded Gaussian kernel:

$$\mathbf{A}_{ij} = \exp\left(-\frac{\text{dist}(i,j)^2}{\sigma_{\text{dist}}^2}\right), \qquad \mathbf{A}_{ij} = 0 \text{ if } \mathbf{A}_{ij} < \epsilon, \tag{36}$$

where $\text{dist}(i,j)$ denotes the road-network distance between sensors $i$ and $j$, $\sigma_{\text{dist}}$ is the bandwidth parameter, and we fix $\epsilon = 0.1$ throughout.

**Evaluation protocols on LargeST.** We consider four evaluation settings:

- **ID forecasting.** We use data from *Jan. 1–Jan. 31, 2019* for training and validation, split chronologically with a 3:1 ratio, and evaluate on a held-out subset of *Feb. 2019* (Table 6).

- **Spatio-temporal OOD generalization.** To simulate a pronounced seasonal distribution shift, we train and validate the model exclusively on *Jan. 2019* (winter) and directly test on *Aug. 2019* (summer), without any adaptation or fine-tuning. We choose August based on a quantitative drift analysis: we compute pairwise Wasserstein distances between the empirical distributions of traffic observations across all months, and visualize the resulting month-to-month distance matrices together with drift trends for **SD**, **CA-D3**, **Orange**, and **LA** in Figure 7. Across these regions, August consistently shows the largest discrepancy relative to January under this metric; accordingly, we use August as the OOD test period.

*Table 6.* Dataset statistics and time splits for training, ID test, and OOD test.

| Dataset | Nodes | Freq | Training & Validation | ID Task Test | OOD Task Test |
|---------|-------|------|------------------------|---------------|----------------|
| **CA-D3** | 480 | 5 min | Jan. 1 – Jan. 31, 2019 | Feb. 1 – Feb. 8, 2019 | Aug. 1 – Aug. 8, 2019 |
| **SD** | 716 | 5 min | Jan. 1 – Jan. 31, 2019 | Feb. 1 – Feb. 8, 2019 | Aug. 1 – Aug. 8, 2019 |
| **Orange** | 953 | 5 min | Jan. 1 – Jan. 31, 2019 | Feb. 1 – Feb. 8, 2019 | Aug. 1 – Aug. 8, 2019 |
| **LA** | 1728 | 5 min | Jan. 1 – Jan. 31, 2019 | Feb. 1 – Feb. 8, 2019 | Aug. 1 – Aug. 8, 2019 |

- **Few-shot cross-city transfer.** We pre-train on source cities (**SD**, **Orange**, **LA**) and adapt to the target region (**CA-D3**) under a strict data budget: only the first $10\%$ time steps of the target-domain training window are used for few-shot fine-tuning, and the remaining target data are reserved for evaluation.

- **Continual forecasting.** We further evaluate continual spatio-temporal forecasting with evolving topology on **PEMS-Stream** (details below), where the model must adapt to newly introduced nodes while mitigating catastrophic forgetting on previously observed nodes.

**PEMS-Stream (Streaming traffic benchmark).** PEMS-Stream is a long-horizon traffic stream spanning *2011–2017*. Following prior continual forecasting practice, we partition the data by year into a sequence of tasks $\{M_m\}_{m=1}^{7}$. The network topology expands over time as new sensors are deployed, with the number of nodes growing from $N{=}655$ to $N{=}871$ (Table 5). Models must incrementally learn new yearly dynamics while preserving performance on previously observed nodes and periods, making PEMS-Stream a stringent benchmark for replay-free continual spatio-temporal modeling.

### F.2. Baseline Details

**Overview.** To comprehensively evaluate the effectiveness of STPDE, we benchmark against a broad suite of state-of-the-art spatio-temporal forecasting methods spanning *ID forecasting*, *spatio-temporal OOD generalization*, *pre-training and transfer*, and *continual forecasting*. Unless otherwise noted, all baselines are implemented using either their *official open-sourced codebases* or a unified re-implementation under **BasicTS** (Liang et al., 2022), and are *tuned on the target dataset* with consistent training and validation splits to ensure a fair comparison.

**General spatio-temporal GNNs (General STGNNs).** These methods primarily assess conventional spatio-temporal forecasting performance under the i.i.d. assumption. Representative architectures include: *GWNet* (Wu et al., 2019), which augments dilated causal convolutions (WaveNet-style) with an adaptive adjacency learning mechanism to capture latent spatial dependencies without a pre-defined graph; *STID* (Shao et al., 2022a), an efficient MLP-based baseline that leverages spatio-temporal identity embeddings to attain strong performance with minimal overhead; $D^2STGNN$ (Shao et al., 2022c), which explicitly decouples spatio-temporal signals into an intrinsic component and a diffusion-like component via gating to separately model deterministic trends and stochastic perturbations; *STAEformer* (Liu et al., 2023a), which incorporates spatio-temporal adaptive embeddings into a Transformer to learn time-step- and node-specific dynamics; *HimNet* (Dong et al., 2024), a hierarchical multi-scale architecture with temporal down-sampling and scale-wise decoupling tailored for long-horizon forecasting; *PatchSTG* (Fang et al., 2025), which brings time-series patching into graph forecasting by constructing patch-level graphs to reduce long-sequence complexity while preserving local semantics; and *BiST* (Ma et al., 2025a), a bilateral architecture with parallel spatio-temporal branches and a correlative memory module that balances accuracy and low-latency inference.

**OOD generalization methods.** These approaches are designed to improve robustness under non-stationary distribution shifts (e.g., seasonal changes, unexpected events, or evolving road networks). We include: *CauSTG* (Zhou et al., 2023), which addresses structural confounding in spatio-temporal graphs via causal backdoor adjustment to suppress spurious correlations and recover invariant mechanisms; *STONE* (Wang et al., 2024), a disentangled invariant learning framework that decomposes patterns into environment-specific and environment-invariant factors and employs invariant risk minimization-style objectives for robust transfer across temporal environments; and *STOP* (Ma et al., 2025b), which studies the impact of graph structural shift through a theoretical generalization bound and introduces structure-aware regularization to promote robustness.

**Pre-training & transfer methods.** These methods evaluate whether large-scale historical data can be exploited for improved adaptation in few-shot or zero-/low-shot regimes. We consider: *STEP* (Shao et al., 2022b), which introduces a masked

autoencoding (MAE) pre-training paradigm for spatio-temporal forecasting using a dedicated spatio-temporal Transformer backbone (TSFormer); *STD-MAE* (Gao et al., 2024), a spatio-temporally decoupled MAE that uses a two-branch design and asymmetric masking to separately capture spatial correlations and temporal dynamics; *FlashST* (Li et al., 2024), a lightweight prompt-tuning transfer framework that injects learnable prompt vectors encoding spatio-temporal context and distribution cues into a frozen pre-trained backbone for rapid few-shot adaptation; and *CrossST* (Liu & Zhang, 2025), which focuses on cross-city transfer via structural pattern matching, reusing a shared dictionary of local subgraph motifs to map structural traffic patterns from source to topologically distinct target networks.

**Continual learning methods.** These baselines target continual forecasting under streaming data with *evolving graphs* (i.e., dynamically increasing node sets), where the key challenge is to learn new tasks while mitigating catastrophic forgetting. We include: *STKEC* (Wang et al., 2023a), which follows an "expansion-and-consolidation" paradigm and leverages topology-aware knowledge distillation to transfer structured knowledge while accommodating newly introduced nodes; *PECPM* (Wang et al., 2023b), which maintains a dynamic pattern memory and performs pattern expansion and consolidation to reuse past traffic motifs while allocating new capacity for emerging dynamics; *STRAP* (Zhang et al., 2025), which proposes structure-aware replay by prioritizing spatio-temporal subgraphs critical for preserving global topological dependencies; and *EAC* (Chen & Liang, 2025), an evolving adaptive consolidation framework that combines importance-weighted elastic regularization with dynamic network expansion to balance retention and adaptation.

### F.3. Implementation Details

**Model configuration.** STPDE follows a compact single-layer *encode–solve–decode* pipeline. Unless otherwise specified, we use a standard 60-minute history window and a 60-minute forecasting horizon: the input sequence length is $T = 12$ (12 × 5-minute steps) and the prediction length is $T' = 12$. In the latent space, all inputs are projected to a hidden dimension $D = 256$. For the Environment Basis Manifold, we set the number of physical bases to $K = 4$. We adopt a sparse Top-$k$ routing mechanism with $k = 2$, i.e., each node's environment-specific parameters are synthesized by mixing the two most relevant bases. For stochastic environment perturbation, we sample perturbed bases with probability $\rho = 0.2$ during training. The auxiliary load-balancing loss is weighted by $\lambda = 0.1$. For the feed-forward network (FFN), we use an expansion ratio of 2.0 and employ SwiGLU as the nonlinear activation to enhance model expressiveness.

**Training and optimization.** All models are optimized using AdamW with an initial learning rate of $1 \times 10^{-3}$. We use mean absolute error (MAE) as the primary training objective, and add an auxiliary load-balancing loss weighted by $\lambda = 0.1$. We train for up to 300 epochs with a batch size of 64. To mitigate overfitting, we apply early stopping based on validation loss with a patience of 50 epochs. We further apply gradient clipping with a maximum gradient norm of 5.0 for stable optimization.

**Compute and reproducibility.** All experiments are conducted on a Linux server cluster. A representative node is equipped with an Intel Xeon Gold 5220 CPU (2.20GHz) and an NVIDIA Tesla V100 GPU (32 GB VRAM). Our software stack includes Python 3.9, PyTorch 2.1.0, and CUDA 11.8. To ensure statistical significance, each setting of STPDE is repeated with 5 different random seeds, and we report the mean performance. During evaluation, predictions are first de-normalized and then assessed using MAE, RMSE, and MAPE.

## G. Additional Experiments

This appendix reports supplementary results that *extend* the main-text analyses (Section 5) beyond **CA-D3**. In particular, we provide (i) ablations and hyperparameter sensitivity on **SD** and **Orange**, and (ii) qualitative case studies on **SD**, **Orange**, and **LA**. All experimental protocols follow Section 5.

### G.1. Ablation Study

We repeat the ablation protocol in Section 5 (*w/o PDE*, *w/o D*, *w/o E*, *w/o P*) on **SD** and **Orange**, with results shown in Figure 8. First, *w/o PDE* leads to the largest degradation and the gap widens under OOD, consistent with Figure 3 and supporting the main-text conclusion that explicitly modeling dynamics via PDE evolution is crucial for robustness. Second, removing environment-aware components (*w/o D* and *w/o E*) consistently hurts performance, with a stronger impact on OOD, corroborating that the gains come from *environment-conditioned* dynamics rather than additional capacity. Finally, disabling stochastic perturbation (*w/o P*) also degrades generalization, aligning with Section 5 that perturbation regularization helps prevent overfitting to a single regime and improves OOD stability.

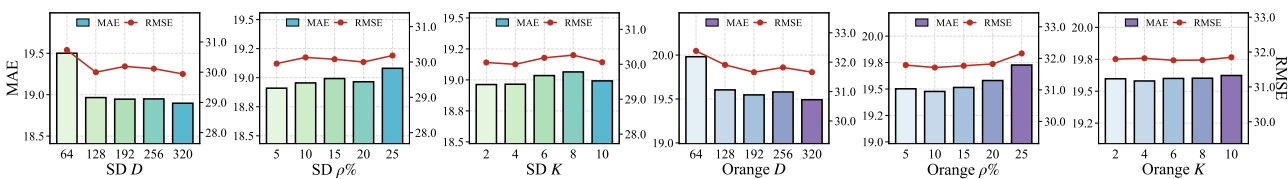

*Figure 8.* Results of ablation experiments on SD and Orange.

*Figure 9.* Results of hyperparameter sensitivity analysis on SD and Orange.

## G.2. Hyperparameter Study

We further conduct the same one-factor sensitivity sweeps as in Section 5 on **SD** and **Orange** (Figure 9). Overall, the trends match the main-text observations: increasing the hidden dimension $D$ improves performance up to a moderate regime and then saturates; the perturbation ratio $\rho$ exhibits a clear bias–variance trade-off (too small weakens regularization, too large introduces excessive noise); and performance is robust to the number of environment bases $K$, where compact basis sets already work well. Notably, the default setting in Section 5 lies in a stable region on both datasets, indicating that the reported gains do not rely on brittle tuning.

## G.3. Case Study

Following the visualization in Section 5, we present additional qualitative results on **SD**, **Orange**, and **LA** in Figure 10. Across these regions, the learned *Environment Basis Manifold* exhibits clear specialization rather than collapsing to a single mode, and the induced routing and assignment patterns remain structured and spatially coherent. These observations are consistent with Section 5: the improved OOD robustness of STPDE is accompanied by an interpretable decomposition into shared propagation structure and environment-specific variations.

To further examine whether the learned Environment Basis Manifold captures meaningful real-world structure rather than merely performing abstract clustering, we visualize the learned basis representations and map the corresponding node groups back to geographic space. As illustrated in Figure 11, nodes assigned to the same latent cluster tend to exhibit similar temporal patterns and are often located along the same highway corridor. These observations suggest that the learned environment bases encode spatio-temporal meta-information that is correlated with physical road-network structures, rather than arbitrary low-rank patterns.

## G.4. Scalability under Long-Stream Continual Learning

We further conduct a synthetic long-stream stress test to examine whether expanding the Environment Basis Manifold introduces excessive parameter, memory, or computational overhead. The synthetic stream contains 20 incremental periods, with the number of nodes increasing from 500 to 10,000. The synthetic data are generated to follow the distributional characteristics of PEMS-Stream. During continual learning, STPDE keeps the invariant backbone frozen and only expands the environment-specific manifold to accommodate newly introduced nodes.

As shown in Figure 12, the number of parameters increases approximately linearly with the incremental stages, while training time, inference time, and GPU memory remain within a moderate range. This suggests that the proposed expansion strategy introduces only lightweight environment-specific parameters, instead of repeatedly enlarging or fine-tuning the full model backbone. Therefore, STPDE remains scalable under long-stream continual learning with dynamically growing graphs.

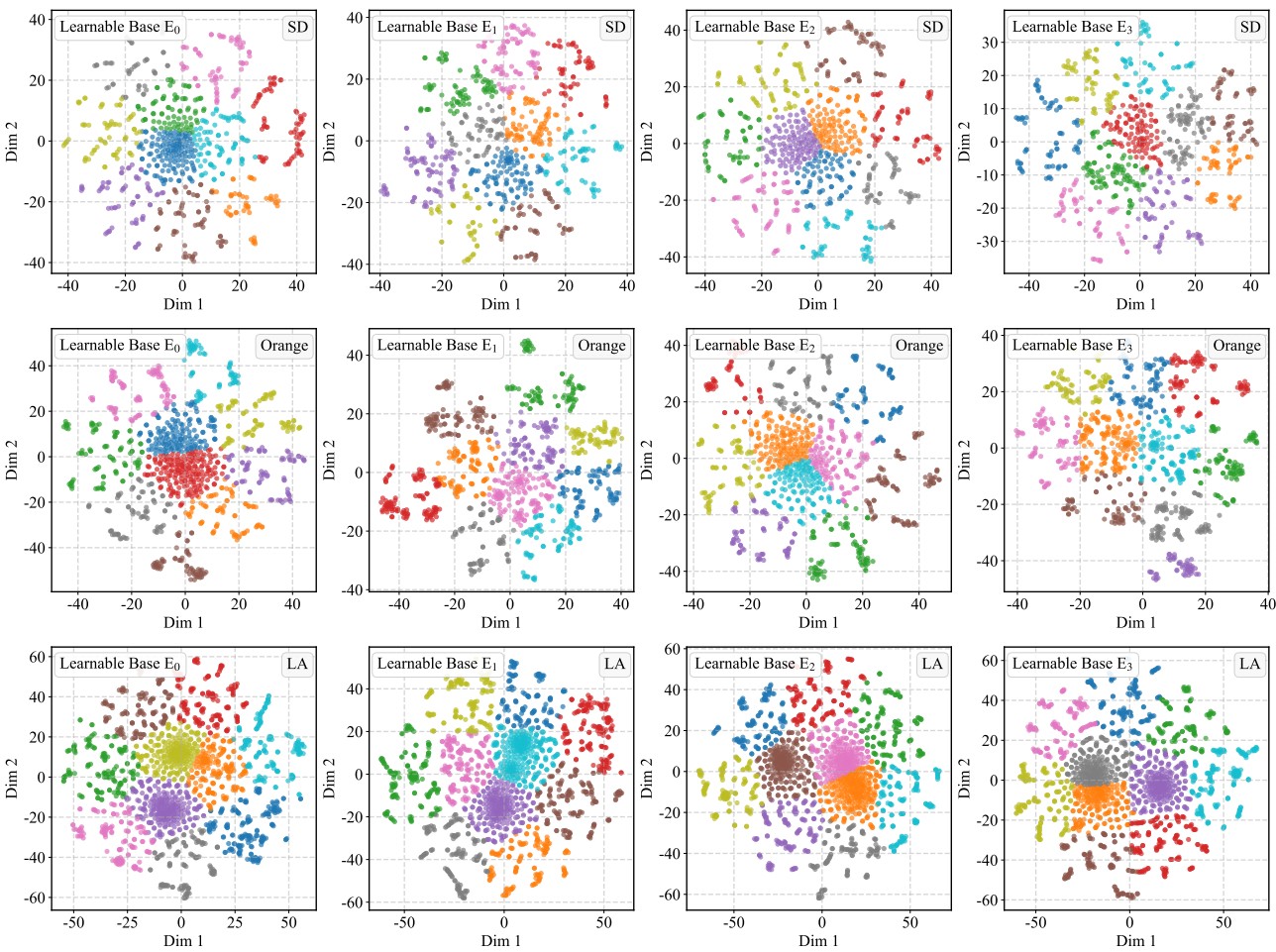

*Figure 10.* Qualitative case study on SD, Orange, and LA.

## G.5. Current Limitations and Future Work

While `STPDE` establishes a principled framework for disentangling spatio-temporal dynamics from environmental hetero-geneity, a deeper physics-oriented inspection reveals several limitations in its modeling assumptions and operator choices. These limitations naturally suggest promising directions for future research.

**Beyond linear diffusion operators.** The current invariant operator in `STPDE` is grounded in thermodynamic diffusion theory and linear Green's functions. This linear approximation is computationally attractive and effective for smooth energy transport, yet it may be insufficient for capturing higher-order dynamics dominated by strong nonlinearity, such as advection- or reaction-driven phenomena in chaotic systems (e.g., traffic shock waves or extreme meteorological turbulence). A compelling future direction is to explore *nonlinear operator learning* that can explicitly represent nonlinear interactions in complex fluid-like dynamics while retaining the favorable computational efficiency of the current solver.

**Adaptive environment basis for long-tail and drifting media.** `STPDE` assumes that infinitely diverse local environments can be reconstructed from a finite and static set of physical bases (i.e., a fixed-size Environment Basis Manifold). Although effective under standard conditions, a static dictionary may become a representational bottleneck when facing long-tailed anomalous environments or non-stationary media whose properties drift over time. Future work may investigate *dynamic and adaptive environment representations* that allow the model to adjust, refine, or expand its physical bases based on real-time observations, potentially strengthening generalization under zero-shot transfer or abrupt environmental changes.

**Continuous-time solvers beyond "time-as-features."** To maximize parallel inference efficiency, `STPDE` adopts a dis-cretized design that transforms temporal dependencies into high-dimensional feature projections, effectively treating "time-as-features." While this paradigm avoids error accumulation from autoregressive rollouts, it may sacrifice fine-grained

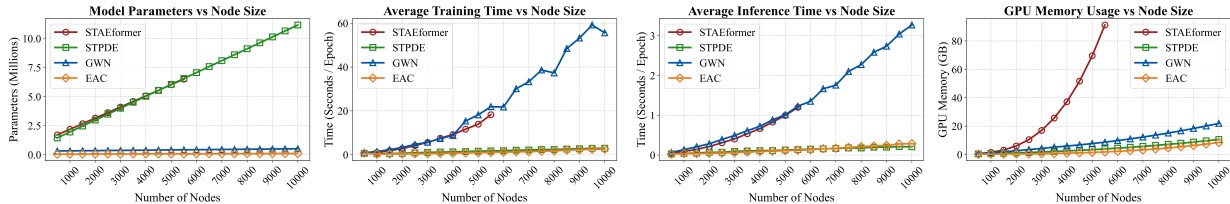

*Figure 11.* Physical interpretation of the learned Environment Basis Manifold. The learned clusters correspond to nodes with similar temporal patterns and coherent geographic locations.

*Figure 12.* Scalability analysis under long-stream continual learning. STPDE is evaluated on a synthetic 20-period stream with the number of nodes growing from 500 to 10,000. The results show that manifold expansion leads to approximately linear parameter growth while maintaining moderate computational overhead.

modeling of physical continuity and reduce flexibility under irregular sampling or missing time steps. An important future direction is to develop a more general spatio-temporal solver that preserves *linear* computation while supporting queries and rollouts at *arbitrary time instants*, enabling robust reasoning under non-uniform temporal grids and partially observed sequences.

