# OpenReview forum: "Decoupling Universal Laws and Environmental Heterogeneity: A Physics-Inspired Framework for Robust Spatio-Temporal Forecasting"
_ICML.cc/2026/Conference — ICML 2026 regular_

### Official Review · Reviewer_1Ngv · 2026-02-23

**Soundness:** 3
**Presentation:** 3
**Significance:** 3
**Originality:** 2
**Overall Recommendation:** 4
**Confidence:** 2

**Summary:**

This paper presents STPDE, a physics-inspired framework designed for robust spatio-temporal forecasting under non-stationary environments and distribution shifts. The authors move away from discrete message-passing paradigms, instead reformulating spatio-temporal dynamics as the numerical evolution of an inhomogeneous partial differential equation (PDE).

**Compliance With Llm Reviewing Policy:**

Affirmed.

**Final Justification:**

The author addressed my concerns; therefore, I raised my score.

**Key Questions For Authors:**

1. Can you provide empirical evidence—such as a visualization of the learned latent field $u(x, \tau)$ or a conservation law analysis—demonstrating that the model captures traffic-specific dynamics that a standard, non-physics-inspired Graph Convolutional Network (GCN) or Transformer cannot?
2. Why was a simple MLP chosen as the primary baseline for this ablation instead of a high-performing non-physical spatio-temporal model like STAEformer or D2STGNN? Furthermore, how do you decouple the performance gains of the PDE framework from the inherent strengths of the linear attention mechanism used to implement it?
3. Given that the number of bases $K$ plateaus at a very low value (typically $K=4$), how do you ensure this isn't simply a low-rank approximation or a clustering trick? Specifically, does the learned manifold correlate with actual physical metadata (e.g., road types, weather, or sensor density)?
4. Have you conducted a "stress test" on the memory and parameter overhead of this expansion over a much longer task stream than the 7 periods in PEMS-Stream?

**Limitations:**

yes

**Strengths And Weaknesses:**

Strengths:
1. Unlike traditional spatio-temporal graph neural networks (STGNNs) that often rely on fixed or learned graph structures that absorb dataset-specific biases, STPDE reformulates forecasting as the evolution of an inhomogeneous partial differential equation (PDE). This allows the model to explicitly separate universal, domain-invariant dynamics (captured by an Invariant Diffusion Operator) from local environment-dependent heterogeneity (parameterized by an Environment Basis Manifold). This theoretical approach provides a principled way to maintain high accuracy in standard settings while remaining robust to non-stationary environments.

Weaknesses:
1. Conceptual Dilution of "Physics": The core premise—modeling traffic as an inhomogeneous partial differential equation (PDE)—is arguably more of a branding exercise than a rigorous physical derivation. Traffic flow involves complex human behavior and non-linear shocks that a simple diffusion-based Laplacian operator fails to capture.
2. Tautological Contribution: Theorem 4.1 claims to prove that linear attention approximates Green's function propagation. This mathematical equivalence is well-established in kernel theory; presenting it as a novel "Theory-Reality Bridge" lacks original depth.
3. Efficiency Misrepresentation: While the paper touts $O(ND^2)$ complexity, the hidden constant factors in the "aggregate-distribute" procedure and high latent dimensions ($D=256$) may negate real-world speed advantages on smaller graphs.

---

> ### Author Rebuttal · Authors · 2026-03-30
>
> We sincerely thank you for the constructive comments.
>
> ---
> > **W1. Physical Motivation of the PDE Framework**
>
> Please allow us to clarify: We do not introduce PDEs for branding purposes, but rather to **use a physically inspired perspective to guide the model design**. Specifically, Eq. 1. allows us to explicitly decouple shared propagation dynamics from environment-specific heterogeneity.
>
> While the integral operator uses a linear diffusion approximation, **STPDE is highly nonlinear**. Environment-conditioned modulation through $\alpha(\Phi,\xi)$ and $S(\Phi,\xi)$, combined with nonlinear adaptation and decoding, allows the model to capture nonlinear patterns and adapt to different environments.
>
> > **W2. Novelty of Theorem 4.1**
>
> The main contribution of this paper is introducing heat diffusion theory as a structured guide for ST modeling to build a robust framework. In this context, Theorem 4.1 is not a new theoretical result; rather, it helps explain how the Invariant Diffusion Operator can be implemented using linear attention.
>
> Although this operator could be approximated with other message-passing methods like GCNs, those usually use more computation and memory on large datasets. So, we use linear attention for efficiency, and **Theorem 4.1 offers a theoretical explanation for this choice**.
>
> > **W3. Model Efficiency on Small Graphs**
>
> We added an efficiency experiment on a small graph dataset to validate our method. As shown in the table below, even on **small graphs** like CA-D5 (211 nodes, D=256), STPDE **maintains a speed advantage** over lightweight models such as GWNet, thanks to its efficient design, while avoiding unnecessary computational cost.
> | Model | Train (s) | Infer (s) | GPU (MiB) |
> |---|---:|---:|---:|
> | GWNet | 14.51 | 2.16 | 1362 |
> | D2STGNN | 44.46 | 6.55 | 5974 |
> | STAEformer | 32.32 | 4.59 | 5466 |
> | STPDE | 9.48 | 2.18 | 1364 |
>
> > **Q1. Evidence of Traffic-Specific Dynamics**
>
> Certainly. We provide evidence from latent-feature visualizations that STPDE captures traffic-specific dynamics beyond those learned by standard non-physics-inspired models. We visualized the pre-decoding latent features of ID and OOD data via t-SNE ([Anonymous Fig. 1](https://anonymous.4open.science/r/Re_1852)).
>
> Fig. 1c shows that a vanilla Transformer produces mixed features, mainly capturing general patterns. Using only the Environment Basis Manifold as a contrasting case (Fig. 1b) leads to overly separated clusters dominated by heterogeneity. In contrast, the full STPDE (Fig. 1d) achieves a balanced representation, **capturing both heterogeneity and invariance**.
>
> > **Q2. Concern About Ablation Study**
>
> Table 1 already includes strong baselines like D2STGNN and STAEformer, but ablating the PDE framework with a simple MLP is insufficient. To address this, we designed **a stronger "w/o PDE"** variant: it uses the same inputs as STAEformer, concatenates spatial embeddings and applies linear graph attention for forecasting (see table below). Results show that the model without PDE guidance performs worse, further demonstrating the importance of the PDE-driven design in STPDE. We will revise this part accordingly.
> | Model | CA-D3 OOD MAE | SD OOD MAE |
> |---|---:|---:|
> | D2STGNN | 17.80 | 21.68 |
> | STAEformer | 17.82 | 22.20 |
> | w/o PDE | 18.75 | 21.72 |
> | STPDE | 16.34 | 19.89 |
>
> > **Q3. Physical Meaning of the Learned Manifold**
>
> To verify that our model captures actual physical metadata, we further visualized the Environment Basis Manifold. Applying t-SNE clustering to the first basis ([Anonymous Fig. 2](https://anonymous.4open.science/r/Re_1852)) reveals that nodes within the same cluster share similar temporal patterns. Furthermore, mapping their geographical locations showed they are situated along the exact same highway. This confirms that the learned manifold successfully captures **meta-information** highly correlated with real-world physical attributes.
>
>
> > **Q4. Stress Test Under Longer Streams**
>
> Since long-horizon real-world streaming data are lacking, we therefore built a synthetic stress-test dataset with 20 incremental periods and a dynamically growing node set (500–10,000 nodes) to compare STPDE against representative baselines. To mimic real-world scenarios as closely as possible, these datasets were generated based on the data distribution of PEMS-Stream.
>
> As shown in [Anonymous Fig. 3](https://anonymous.4open.science/r/Re_1852), STPDE's parameters grow linearly with the incremental stages as the Environment Basis Manifold expands to encode new nodes, and this lightweight expansion **does not incur excessive computational overhead**. Even under this extreme stress test, STPDE maintains minimal training and inference times and a low memory footprint, demonstrating robust scalability.
>
> ---
> All supplementary experiments will be included in the Appendix. We are happy to answer any further questions and would appreciate your consideration in revising the score.

---

> > ### Author Rebuttal · Reviewer_1Ngv · 2026-04-01
> >
> > The authors have addressed my concerns regarding the model's motivation and evaluation. I am satisfied with the response and will increase my score to Weak Accept.

---

> > > ### Author Response · Authors · 2026-04-01
> > >
> > > Dear Reviewer,
> > >
> > > Thank you for your time and effort in reviewing our manuscript and rebuttal. We are grateful for your recognition and willingness to reconsider your score, and we will carefully apply your suggestions in our future work.  We would deeply appreciate it if you might consider raising your confidence score.
> > >
> > > Sincerely,
> > >
> > > The Authors

---

### Official Review · Reviewer_xSPt · 2026-03-12

**Soundness:** 2
**Presentation:** 2
**Significance:** 2
**Originality:** 2
**Overall Recommendation:** 3
**Confidence:** 4

**Summary:**

This paper proposes STPDE, a physics-inspired framework for ST forecasting that aims to improve robustness under distribution shifts. The core idea is to reinterpret ST prediction as the numerical evolution of an inhomogeneous partial differential equation (PDE). The proposed framework decomposes the dynamics into two components: an Invariant Diffusion Operator, which captures universal propagation laws across environments, and an Environment Basis Manifold, which models environment-specific heterogeneity through sparse routing and stochastic perturbations. STPDE follows an encode-solve-decode pipeline, where historical observations are encoded into latent states, evolved via a neural PDE solver, and decoded into future predictions. The authors evaluate STPDE on traffic benchmarks under several settings including in-distribution forecasting, out-of-distribution generalization, few-shot cross-city transfer, and continual learning.

**Compliance With Llm Reviewing Policy:**

Affirmed.

**Final Justification:**

In my final assessment, I acknowledge that the paper has clear merits: the physics-inspired formulation is interesting, the overall motivation is meaningful, and the authors provided a thorough rebuttal with additional ablations, efficiency analysis, and cross-domain results. These clarifications partially addressed my concerns and improved the presentation of the work. However, they did not fundamentally change my evaluation. I remain unconvinced that the PDE-based decomposition and the added conceptual/architectural machinery provide sufficiently strong practical advantages over simpler strong baselines or existing pretraining/foundation-model style paradigms. In particular, some improvements are still modest or scenario-dependent, and the current empirical evidence does not fully justify the strength of the paper’s broader claims regarding necessity, robustness, and general advantage. Overall, I view the work as promising but still leaving substantial room for improvement in validation and positioning, so I am keeping my score unchanged.

**Key Questions For Authors:**

Q1: The framework introduces several interacting modules (PDE solver, invariant diffusion operator, environment manifold, perturbation). Can the authors evaluate which components are essential and whether similar performance could be achieved with a simpler architecture?

Q2: The paper claims that attention approximates the Green’s function of the diffusion equation. Is there empirical evidence that the learned attention kernels actually behave like diffusion kernels, or is this primarily a conceptual analogy?

Q3: The proposed model is more complex than many existing spatio-temporal forecasting architectures. How does its training stability and hyperparameter sensitivity compare with simpler baselines in practice?

Q4: The evaluation focuses on traffic forecasting benchmarks. Does the PDE-inspired formulation generalize to other spatio-temporal domains (e.g., climate or environmental systems)?

**Limitations:**

L1: The proposed framework introduces multiple conceptual and architectural components, which significantly increases model complexity and may hinder reproducibility and practical application.

L2: The theoretical justification relies heavily on physical analogies (e.g., diffusion processes and Green's functions), but the empirical validation of these assumptions should be further emphasized in the paper.

L3: The experimental improvements over strong baselines appear relatively small in some settings, making it unclear whether the proposed complexity is necessary.

L4: The evaluation is limited to traffic forecasting datasets, which raises questions about the general applicability of the method to other spatial-temporal prediction tasks, such as climate/weather, environmental system, and earth system forecasting.

**Strengths And Weaknesses:**

### Strengths

**Technical soundness and interpretability:** The paper introduces a physics-inspired formulation based on partial differential equations (PDEs) to model ST dynamics. From a theoretical standpoint, this formulation is reasonable and provides a clear interpretation of the modeling process. By explicitly linking diffusion processes and spatial-temporal propagation, the framework offers a principled perspective that goes beyond purely heuristic neural architectures.

---

### Weaknesses

- The overall architecture of STPDE introduces multiple interacting components, including the PDE solver, invariant diffusion operator, environment basis manifold, stochastic perturbations, and routing mechanisms. While each component is individually motivated, the combined system appears quite complex, which may hinder practical deployment, reproducibility, and adoption in real-world applications.

- **Limited ability of basic mathematical models to capture real-world complexity:** In realistic spatial-temporal systems, the factors influencing predictions are often highly complex and heterogeneous, including infrastructure conditions, weather effects, human mobility patterns, policy interventions, and unexpected events. Modeling such diverse influences using a relatively structured mathematical formulation (e.g., PDE-based diffusion dynamics) may be restrictive. In contrast, highly expressive deep learning models are often better suited to capture such complex and nonlinear interactions.

- **Related ideas have already been explored through other paradigms:** The key idea of separating universal patterns from environment-specific characteristics has also been addressed in other popular approaches. For example, **pre-training and fine-tuning paradigms** can learn generalizable representations from large datasets while adapting to new environments through lightweight fine-tuning. Similarly, recent **Spatial-Temporal Foundation Models** aim to capture general propagation patterns and adapt them across tasks and domains. Therefore, it is unclear whether the proposed PDE-based decomposition provides substantial advantages compared to these existing paradigms.

---

> ### Author Rebuttal · Authors · 2026-03-30
>
> We sincerely thank you for the constructive comments.
>
> ---
> > **W1. & Q1. Model Complexity and Component Necessity**
>
> We would like to clarify that **STPDE is not a complex model**—combining PDE theory with neural networks may make it seem that way. Its core is a lightweight Physics-Inspired PDE Solver with two main components: the Invariant Diffusion Operator (D) and the Environment Basis Manifold (E) with perturbations (P).
>
> Ablation studies confirm that **each component is important**: on the CA-D3 dataset, MAE increases by 32.54% without D, 17.95% without E, and 15.32% without P. Impressively, the whole core is implemented in just 200 lines of code. Rather than being complex, STPDE is **easy to deploy, scales well, and handles large networks efficiently** across different ST scenarios.
>
> > **W2. Ability to Capture Real-World Complexity**
>
> Please allow us to clarify: **STPDE is still a highly expressive deep learning model**. We introduce PDEs not to simulate the real world with a closed-form physical model, but to provide a strong structural guide for designing the network.
>
> For complex heterogeneous factors, we do not rely on simple analytical formulas. Instead, both the diffusion coefficient $\alpha(\Phi,\xi)$ and the source term $S(\Phi,\xi)$ are deeply parameterized. In particular, the Environment Basis Manifold uses nonlinear neural mappings to adaptively encode these factors into **highly nonlinear** spatial modulation coefficients and source terms.
>
> In short, STPDE combines a PDE-inspired decomposition into physically meaningful components with neural networks that capture micro-level nonlinear complexity.
>
> > **W3. Advantages Over Existing Paradigms**
>
> The core idea of separating universal patterns from environment-specific characteristics is not unique. What sets STPDE apart is that it makes this separation more **interpretable** through theory-guided decomposition, while also achieving a **more effective** separation mechanism.
>
> In experiments, STPDE outperforms pre-training methods such as FlashST and CrossST, and we additionally compare it with the ST foundation model OpenCity. In few-shot cross-city transfer on CA-D3, STPDE achieves an MAE of 16.25, compared with FlashST (18.49), CrossST (17.51), and OpenCity (20.58). These results demonstrate that the physics-inspired separation approach in STPDE is both more effective and interpretable.
>
>
> > **Q2. Diffusion Interpretation**
>
> To address this question, [Anonymous Fig. 4](https://anonymous.4open.science/r/Re_1852) provides three pieces of evidence that the learned attention kernels exhibit diffusion-like behavior.
>
> First, in Fig. 4(a), the attention weights show a **clear log-linear decay** with shortest-path distance on the road network, suggesting that nearby locations interact more strongly than distant ones.
>
> Second, in Fig. 4(b), we compare the learned attention matrix with the analytical heat kernel $\exp(-L)$ derived from the road-network Laplacian $L$. The two show a **statistically significant positive correlation** (Pearson $r = 0.0575, p < 0.001$), indicating that the learned kernel is structurally aligned with the physical Green’s function.
>
> Third, in Fig. 4(c), when projecting the attention weights of a hub node onto real geographic coordinates, high-weight regions **spread outward** along connected road paths rather than forming isolated jumps.
>
> Overall, these quantitative and visual results suggest that the attention mechanism in STPDE is not just a conceptual analogy but actually captures physics-aware continuous propagation.
>
> > **Q3. Training Stability and Hyperparameter Sensitivity**
>
> We provide additional loss curves on CA-D3 and SD in [Anonymous Fig. 5](https://anonymous.4open.science/r/Re_1852) . STPDE converges more **smoothly and faster** than D2STGNN and STAEformer, with fewer oscillations, which is due to its compact design and the regularization effect of the physics-inspired structure.
>
> STPDE is also easy to tune: its main hyperparameters are only $D, K, \rho%$, and the model remains **stable** across a wide range of settings. Overall, the PDE prior does not make training harder and improves robustness in practice.
>
> > **Q4. Cross-Domain Generalization**
>
> We further evaluate STPDE on the open air-quality forecasting dataset **AIR-BTHSA** [1], including its OOD setting. As shown in the table below, STPDE achieves the best overall performance, indicating that the framework generalizes beyond traffic forecasting to other ST systems.
>
> | Model | AIR-BTHSA MAE | AIR-BTHSA OOD MAE |
> |---|---:|---:|
> | GWN | 40.86 | 27.00 |
> | STID | 17.57 | 20.91 |
> | D2STGNN | 16.65 | 20.96 |
> | STAEformer | 16.67 | 20.75 |
> | CauSTG | / | 20.03 |
> | STPDE | 15.87 | 19.51 |
>
> [1] Wang, Shuo, et al. "KnowAir-V2: A Benchmark Dataset for Air Quality Forecasting with PCDCNet."
>
> ---
> All supplementary experiments will be included in the Appendix. We are happy to answer any further questions and would appreciate your consideration in revising the score.

---

> > ### Author Rebuttal · Reviewer_xSPt · 2026-04-03
> >
> > The authors provide helpful clarifications and additional results. Nevertheless, the rebuttal does not fully resolve my concerns about the overall architectural complexity, the marginal improvements over strong baselines, and the practical advantages over existing paradigms, so my evaluation remains unchanged.

---

> > > ### Author Response · Authors · 2026-04-04
> > >
> > > Dear Reviewer XSPt,
> > >
> > > We sincerely thank the reviewer for the thoughtful feedback and for acknowledging our clarifications. We understand your remaining concerns and would like to address them directly with concrete evidence:
> > >
> > > ---
> > > > **1. On Architectural Complexity**
> > >
> > > While the PDE perspective introduces new theoretical concepts, the actual implementation is highly streamlined. STPDE is a single-pass `encode-solver-decode` framework. The core solver relies only on an invariant diffusion operator (using linear attention to ensure $\mathcal{O}(ND^{2})$ complexity) and an environment basis manifold. Our ablations prove these are strictly necessary, not redundant: removing D, E, or P increases MAE on CA-D3 by 32.54%, 17.95%, and 15.32%, respectively.
> > >
> > > Furthermore, on the large-scale LA dataset, STPDE's computational cost demonstrates a massive advantage: **compared to the standard strong baseline GWNet and the recent lightweight model PatchSTG, STPDE not only reduces MAE by 6.1% and 13.5% respectively, but also cuts training time by 79.3% and 51.1%, while saving 63.1% and 48.4% in GPU memory.** The "complexity" lies entirely in conceptual clarity, not in deployability or computational overhead.
> > >
> > > > **2. On "Marginal" Improvements Over Strong Baselines**
> > >
> > > STPDE is purposefully optimized to tackle **distribution shifts**, not merely to marginally push stationary ID benchmarks. Even when compared against the latest SOTA models (e.g., **PatchSTG [KDD 2025], BiST [VLDB 2025], STOP [ICML 2025], EAC [ICLR 2025], and STRAP [NeurIPS 2025]**), STPDE's performance gains compound significantly as the scenario difficulty increases:
> > > * **ID forecasting:** Error reduced by **1.5%**
> > > * **OOD generalization:** Error reduced by **4.0%**
> > > * **Few-shot cross-city transfer:** Error reduced by **7.2%**
> > > * **Continual learning:** Error reduced by **27.2%**
> > >
> > > > **3. On Practical Advantages Over Existing Paradigms**
> > >
> > > Compared to existing pre-training paradigms and foundation models, STPDE's practical advantage lies in its **explicit, structured decoupling**. The invariant operator locks in shared dynamics, while the lightweight environment manifold enables rapid adaptation, yielding superior data efficiency. In few-shot cross-city transfer (CA-D3), STPDE (MAE **16.25**) clearly outperforms FlashST (**18.49**), CrossST (**17.51**), and the foundation model OpenCity (**20.58**). Moreover, cross-domain validation on the AIR-BTHSA air quality dataset (ID: 15.87 / OOD: 19.51) confirms it surpasses STID, D2STGNN, and STAEformer.
> > >
> > > These results demonstrate that STPDE offers tangible engineering benefits in the accuracy-efficiency-generalization trade-off, rather than just being "another way to explain things."
> > >
> > > ---
> > >
> > > We again sincerely appreciate your time and feedback. We would be very grateful if you could kindly reconsider your current score.
> > >
> > > Sincerely,
> > >
> > > The Authors

---

### Official Review · Reviewer_feNw · 2026-03-13

**Soundness:** 4
**Presentation:** 3
**Significance:** 3
**Originality:** 4
**Overall Recommendation:** 5
**Confidence:** 4

**Summary:**

This paper addresses the generalization challenge of spatio-temporal forecasting models under non-stationary environments and proposes an inhomogeneous PDE-based STPDE framework. By decoupling invariant diffusion dynamics from environmental heterogeneity, STPDE significantly improves model robustness. The paper validates its effectiveness across multiple forecasting settings and achieves consistent performance gains over mainstream methods.

**Compliance With Llm Reviewing Policy:**

Affirmed.

**Final Justification:**

Based on the comprehensive and targeted response, my concerns about the methodology, evaluation, and experiments have been fully addressed. The response further enhances my confidence in its quality and technical soundness. I will maintain my score of 5.

**Key Questions For Authors:**

The questions are as follows:

Q1. In ID forecasting, why is the error reduction of STPDE not significant on some datasets? Does this suggest that the decoupling mechanism can be redundant in fully stationary environments?

Q2. In the inhomogeneous diffusion equation, could learning the diffusion coefficient $\alpha$ without explicit physical constraints lead to training instability?

Q3. In continual learning, adapting to new nodes requires dynamically expanding the environment basis manifold. As the number of tasks increases, could this expansion cause excessive growth in the number of model parameters?

**Limitations:**

Yes

**Strengths And Weaknesses:**

The strengths and weaknesses are as follows:

S1. Unlike purely data-driven black-box models, this work formulates spatio-temporal dynamics as the evolution of an inhomogeneous PDE, providing a physically interpretable solution to distribution shift in spatio-temporal forecasting.

S2. The decoupling between invariant dynamics and environmental heterogeneity is practical. In cross-city transfer and continual learning, the model can adapt to new environments by only fine-tuning the environment basis manifold, substantially reducing transfer training cost and data demand.

S3. The evaluation is thorough, covering standard forecasting, few-shot transfer, and continual learning to verify robustness. The paper further supports each core component with detailed ablations, hyperparameter analysis, visualizations, operator-theoretic derivations, and open-sourced code.

W1. In in-distribution settings, STPDE does not achieve the best performance on all metrics across all datasets (e.g., metrics on the SD and Orange dataset).

W2. The paper lacks comparisons with existing PDE-based or physics-driven spatio-temporal forecasting methods.

---

> ### Author Rebuttal · Authors · 2026-03-31
>
> We sincerely thank you for the constructive comments.
>
> ---
>
> > **W1. & Q1. ID Performance and the Necessity of Decoupling**
>
> In relatively stable environments, some scenario-specific models may perform better than STPDE, since they are tailored for those conditions. However, STPDE is designed for more challenging situations, like OOD generalization, cross-city few-shot transfer, and continual learning, where separating shared patterns from environment-specific differences is crucial.
>
> **This separation is still useful in stable environments**: the ablation study in Fig. 3 shows that removing PDE modeling or key components reduces performance even in standard ID settings. Overall, STPDE handles both stable and changing scenarios well, with its advantages being especially clear in the more complex cases. Compared with the strongest baseline, STPDE improves performance by **4.0%, 7.2%, and 27.2%** in OOD generalization, cross-city few-shot transfer, and continual learning, respectively.
>
> > **W2. Comparisons With Existing Physics-Driven Methods**
>
> We would like to clarify that existing PDE-based or physics-driven spatio-temporal forecasting methods **remain limited**, and many lack directly reproducible, well-maintained open-source implementations for fair comparison. Moreover, most are designed for ID settings, and their predictive performance often falls short of strong baselines such as STAEFormer and D2STGNN.
>
> In contrast, our focus extends beyond ID forecasting to more challenging scenarios, including OOD generalization, cross-domain few-shot transfer, and continual learning. Thus, existing methods are not fully aligned with our task settings. In the revision, we will add a detailed discussion to clarify the distinction.
>
> > **Q2. Stability of Learning Diffusion Coefficients**
>
> Directly learning an unconstrained diffusion coefficient for every node could indeed cause training instability. However, STPDE does not operate this way. Instead, environment-related parameters are generated in a structured manner via the Environment Basis Manifold. Shared environment bases represent different environments, and sparse routing combined with AdaLN assigns suitable combinations to nodes. This reduces the parameter space and adds structure, helping prevent uncontrolled optimization. We also introduce a load-balancing loss to avoid routing collapse, and use AdamW, early stopping, and gradient clipping to further ensure stability. In other words, **the learned diffusion coefficients are not completely free, but environment-specific parameters constrained by shared structure**.
>
> From our experiments, we observe no instability. [Anonymous Fig. 5](https://anonymous.4open.science/r/Re_1852) shows that STPDE’s training loss evolves steadily. In addition, hyperparameter analysis (Section 5.4 and Appendix G.2) demonstrates stable performance across a wide range of perturbation strengths and numbers of environment bases. Performance changes smoothly when bases vary from 2 to 10 or perturbation ratios from 5% to 25%, addressing concerns about instability.
>
>
> > **Q3. Parameter Growth in Continual Learning**
>
> The number of parameters does increase, but this growth is limited to a lightweight environment-specific module rather than the whole backbone. In continual learning, STPDE keeps the shared invariant backbone frozen and only expands and updates the Environment Basis Manifold. This means new parameters are used solely to adapt to new environments and nodes, without inflating the full model, resulting in roughly linear growth.
>
> As shown in [Anonymous Fig. 3](https://anonymous.4open.science/r/Re_1852) , although the parameter count rises with incremental stages, this lightweight expansion does not create significant computational overhead. **STPDE maintains low memory usage and fast training and inference**. Compared with methods that continually expand the main backbone or fine-tune all parameters repeatedly, this approach is much gentler and aligns with our design principle: allowing variability in the environment layer while preserving invariance in the physical layer.
>
> ---
> All supplementary experiments will be included in the Appendix. We are happy to answer any further questions.

---

> > ### Author Rebuttal · Reviewer_feNw · 2026-04-03
> >
> > The additional clarifications and evidence have addressed my previous concerns, including the method's details and robustness in challenging settings. The importance of the decoupling mechanism, along with the model's stability and parameter scalability in extreme scenarios, is now clear. I will keep a positive attitude toward this paper, recommending acceptance.

---

> > > ### Author Response · Authors · 2026-04-03
> > >
> > > Dear Reviewer,
> > >
> > > Thank you for your feedback and for your positive attitude toward our paper. We are glad our clarifications addressed your concerns. We will continue to refine our work accordingly.
> > >
> > > Sincerely,
> > >
> > > The Authors

---

### Official Review · Reviewer_jgKu · 2026-03-14

**Soundness:** 3
**Presentation:** 4
**Significance:** 4
**Originality:** 4
**Overall Recommendation:** 6
**Confidence:** 4

**Summary:**

This paper addresses the sharp performance degradation of spatio-temporal forecasting models under non-stationary environments caused by distribution shift, and proposes a physics-inspired STPDE framework. STPDE reformulates spatio-temporal dynamics as the evolution of an inhomogeneous partial differential equation, explicitly decoupling universal laws from environmental heterogeneity. The method achieves SOTA results on multiple spatio-temporal forecasting tasks, offering a new perspective for handling distribution shift in this area.

**Compliance With Llm Reviewing Policy:**

Affirmed.

**Final Justification:**

The authors have provided a clear and convincing rebuttal that addresses my concerns. The supplementary experiments and detailed explanations effectively address my previous doubts and provide a deeper understanding of the manuscript's contributions. Consequently, I will raise my score.

**Key Questions For Authors:**

See Weaknesses.

**Limitations:**

Yes

**Strengths And Weaknesses:**

Strengths:

- Compared with mainstream methods, using PDE theory of continuous physical fields to guide spatio-temporal dynamics modeling better matches the physical nature of real spatio-temporal systems.

- The proposed Invariant Diffusion Operator effectively approximates the Green’s function via linear attention. This design enables global long-range diffusion modeling while reducing computational complexity to linear, achieving a balance between expressiveness and computational efficiency in large-scale scenarios.

- The Environment Basis Manifold combines sparse routing, stochastic perturbations, and a load-balancing loss. This mechanism enables flexible modeling of local environmental heterogeneity, effectively avoids router collapse, and enhances generalization in OOD scenarios.

- Comprehensive experimental settings covering four scenarios: in-distribution (ID) forecasting, OOD generalization, few-shot cross-city transfer, and continual learning. The paper also provides thorough comparisons with multiple SOTA baselines.

Weaknesses:

-  The model assumes the physical process is dominated by the diffusion equation by default. However, in complex systems such as traffic flow, nonlinear advection terms ($\mathbf{v}\cdot\nabla u$) are also critical to the dynamics. Why does Eq. (1) not consider using an advection-diffusion equation ($\frac{\partial u}{\partial \tau} = \alpha \nabla^2 u - \mathbf{v}\cdot\nabla u + \mathcal{S}$) as theoretical guidance?

- The adopted time-as-features design is beneficial for parallel inference, but it may sacrifice the model’s ability to capture fine-grained continuous-time dynamics.

- The “invariance” emphasized in the paper seems to be achieved more implicitly via architectural constraints. The paper lacks quantitative or visual analysis of the learned operators’ true “invariance” in cross-domain scenarios. Could relevant visualization experiments be added to improve interpretability?

---

> ### Author Rebuttal · Authors · 2026-03-31
>
> We sincerely thank you for the thoughtful comments.
>
>
> ---
>
>
> > **W1. Reason for Not Using an Advection-Diffusion Equation in Eq. 1**
>
> In complex macroscopic systems such as traffic flow, nonlinear advection is also important in addition to diffusion. We have explored related designs—for example, using **GCNs** to capture topology-constrained “local flow” transmission corresponding to the advection term, while using linear attention to model relatively topology-free “global field” interactions corresponding to the diffusion term.
>
> However, explicitly introducing the advection term raises two key concerns. First, one goal of STPDE is strong robustness and cross-scenario generalization, whereas GCNs typically relies on a predefined topology. In complex scenarios, fixed topology can easily **limit generalization and harm prediction performance**. Second, graph convolution usually incurs **higher computational costs** in large-scale spatio-temporal systems, which is unfavorable for efficient deployment.
>
> For these reasons, we do not directly adopt an advection-diffusion equation. Instead, we incorporate advection effects into the diffusion framework in a simplified way by introducing an **optional module** that combines the predefined adjacency matrix $A$ with linear attention to enhance local message passing. This approach supplements local flow information while preserving STPDE’s robustness, scalability, and computational efficiency.
>
> > **W2. Impact of Time-as-Features on Capturing Fine-Grained Temporal Dynamics**
>
> We added additional experiments to verify that combining time-as-features with the PDE framework does not degrade performance. As shown below, when time-as-features is removed and the temporal dimension is used directly, model performance drops substantially. This suggests that without effective temporal modeling, temporal dependencies cannot be fully captured, limiting the model’s capacity.
>
> In contrast, time-as-features encodes temporal information into **hidden node-level representations** that are fed into the PDE solver, allowing temporal dependencies to be modeled implicitly. These results indicate that, compared with the more direct temporal modeling strategies used in existing methods, this implicit design is more effective within our framework.
>
> | Model | CA-D3 MAE | CA-D3 RMSE | SD MAE | SD RMSE |
> |---|---:|---:|---:|---:|
> | STPDE w/o TAF | 18.34 | 30.40 | 21.33 | 33.77 |
> | STPDE | 13.57 | 23.08 | 18.96 | 29.99 |
>
>
>
> | Model | CA-D3 OOD MAE | CA-D3 OOD RMSE | SD OOD MAE | SD OOD RMSE |
> |---|---:|---:|---:|---:|
> | STPDE w/o TAF | 22.43 | 33.55 | 24.34 | 36.93 |
> | STPDE | 16.34 | 24.12 | 19.89 | 30.21 |
>
> > **W3. Improving Interpretability of Invariance via Visualization**
>
> We added visualization analyses of the model under different domains (ID and OOD). Specifically, we extracted latent representations of in-distribution and out-of-distribution samples before the final regression layer and projected them using t-SNE (see [Anonymous Fig. 1](https://anonymous.4open.science/r/Re_1852)).
>
> Our core design is that the invariant diffusion operator captures patterns shared across environments, while the environment basis manifold encodes environment-specific heterogeneity. The visualizations show that using only the invariant diffusion operator produces more **entangled features** that mainly capture general shared patterns, whereas using only the environment basis manifold produces clearly **separated features** that emphasize environment-specific patterns.
>
> In contrast, STPDE **balances the two**: it effectively separates ID and OOD representations while maintaining clear internal structure within each domain. At the same time, some overlap remains, reflecting the model’s ability to extract invariant cross-domain patterns.
>
> ---
> All supplementary experiments will be included in the Appendix. We are happy to answer any further questions.

---

> > ### Author Rebuttal · Reviewer_jgKu · 2026-04-03
> >
> > The rebuttal has addressed all of my concerns. Their detailed explanations provide valuable insights that strengthen my understanding of the manuscript. I will accordingly raise my score.

---

> > > ### Author Response · Authors · 2026-04-03
> > >
> > > Dear Reviewer,
> > >
> > > We sincerely appreciate your feedback and are grateful for your decision to raise your score! We are glad our rebuttal addressed your concerns. We will continue to refine it accordingly.
> > >
> > > Sincerely,
> > >
> > > The Authors

---

### Official Review · Reviewer_mv3M · 2026-03-19

**Soundness:** 3
**Presentation:** 4
**Significance:** 4
**Originality:** 3
**Overall Recommendation:** 5
**Confidence:** 4

**Summary:**

This paper proposes a physics-inspired model named STPDE, which addresses the problem of data distribution shift by reframing spatiotemporal prediction tasks as non-homogeneous partial differential equations (PDEs). Central to this design is the decomposition of system dynamics into an invariant diffusion operator and an environmental base manifold; thanks to this architecture, the proposed method not only possesses a solid theoretical foundation but has also been thoroughly validated and supported by extensive experimental results.

**Compliance With Llm Reviewing Policy:**

Affirmed.

**Final Justification:**

The author addressed my concerns; therefore, I raised my score.

**Key Questions For Authors:**

1. What are the specific advantages and limitations of the PDE-based approach compared with spatio-temporal ODE methods?

2. STPDE unifies multiple tasks. Can it, through large-scale pretraining, attain zero-shot prediction capabilities similar to foundation models such as OpenCity?

3. Could the paper clarify whether the OOD dataset obtained through monthly sampling truly exhibits distribution shift? This needs to be explicitly addressed in the relevant section.

**Limitations:**

The author discussed the limitations in the paper.

**Strengths And Weaknesses:**

- Strengths
1. This paper addresses a significant challenge: unifying four distinct scenarios—In-Domain (ID), Out-of-Domain (OOD), Few-shot, and Continual Forecasting—within a single model framework.

2. The proposed method constitutes a general framework that recasts spatiotemporal dynamics as an evolutionary process governed by non-homogeneous partial differential equations; this approach effectively decouples universal dynamic laws from the heterogeneous characteristics specific to particular environments.

3. The experimental evaluation presented in the paper is comprehensive, featuring comparisons against benchmarks drawn from diverse domains; the extensive experimental results clearly demonstrate the effectiveness of the proposed model.

4. The paper is exceptionally well-presented, with carefully crafted figures and tables that effectively illustrate the key findings.

- Weaknesses

1. Although the environment basis Φ is extendable across scenarios, it essentially consists of trainable weights that remain fixed after the training phase concludes. Given this static nature, how does it possess real-time adaptation capabilities against distribution shifts during the inference phase?

2. While the optional geometric prior matrix A can accelerate convergence in ID scenarios, it may impose constraints when facing OOD shifts. Therefore, further discussion is required regarding the specific impact of this parameter on model performance and convergence speed under both ID and OOD settings.

3. The OOD settings in the paper's experiments are established by sampling data from different months, which essentially reflects only seasonal variations. It remains to be evaluated whether the model can handle OOD scenarios involving non-extreme or sudden abnormal environmental conditions.


- Overall, this manuscript stands out as a high-quality contribution among the papers I have reviewed. Should the authors adequately address the concerns raised above, the paper could be further strengthened and reach an even higher standard.

---

> ### Author Rebuttal · Authors · 2026-03-31
>
> We sincerely thank you for the constructive comments.
>
> ---
> > **W1. Real-Time Adaptation of the Environment Basis**
>
> We would like to clarify that real-time adaptation during inference is essentially an online learning setting, whereas STPDE is not designed for that. The stochastic environmental perturbations introduced during training are meant to expose the model to a wider range of environments, reducing overfitting to a single distribution and improving OOD robustness **without updating parameters at test time**.
>
> We also note in the appendix that a fixed-size, static Environment Basis Manifold could become a bottleneck for rare or continuously changing environments. Future work could explore dynamic, scalable environment representations or ones that adapt based on real-time observations. We also plan to incorporate test-time adaptation techniques to further address this limitation.
>
> > **W2. Impact of the Geometric Prior Matrix $A$ in ID and OOD Settings**
>
> We added further analysis of the geometric prior matrix $A$. The results show that $A$ mainly provides a local structural bias: in ID settings, it can speed up training and give modest gains; however, in OOD settings, a fixed $A$ may add constraints that slightly reduce generalization. This aligns with our design intention: $A$ is an optional local prior, not a primary source of OOD robustness.
>
> | Model | CA-D3 MAE | SD MAE | CA-D3 OOD MAE | SD OOD MAE |
> |---|---:|---:|---:|---:|
> | w_A | 13.57 | 18.96 | 16.66 | 20.43 |
> | STPDE | 13.62 | 19.11 | 16.34 | 19.89 |
>
> > **W3. OOD Settings Based on Monthly Sampling**
>
> Sudden or extreme abnormal conditions also deserve attention. To test this, we created a simulated **abnormal environment** from the original OOD data: we randomly picked 10% of time steps as abnormal, perturbed 10% of nodes at each of these steps, and added Gaussian noise (σ = 1.0) to mimic sudden local disturbances. As shown in the table below, STPDE still outperforms GWNet, D2STGNN, and CauSTG, showing that it handles both seasonal shifts and sudden local shocks.
> | Model | CA-D3 OOD MAE | SD OOD MAE |
> |---|---:|---:|
> | GWNet   | 18.45 | 22.83 |
> | D2STGNN | 18.68 | 23.68 |
> | CauSTG  | 18.52 | 23.12 |
> | STPDE   | 17.98 | 22.40 |
>
> > **Q1. Advantages and Limitations of PDE-Based Methods Compared with ST ODE Methods**
>
> In ST forecasting, ODE-based methods focus on the continuous evolution of states over time, making them suitable for **continuous-time modeling and irregular sampling**. PDE-based methods, in contrast, explicitly capture **the interaction between temporal evolution and spatial propagation**, which is essential for ST fields like traffic flow, involving diffusion, propagation, and spatial heterogeneity.
>
> Compared with ODE-based approaches, PDE-based methods naturally model shared propagation laws and local environmental differences, offering stronger physical interpretability. Their main limitation is less flexibility in handling continuous-time dynamics—for instance, in cases of irregular time intervals, prediction at arbitrary time points, or large changes in temporal resolution, ODE-based methods are often easier to apply.
>
> > **Q2. Potential for Zero-Shot Prediction Through Large-Scale Pretraining**
>
>
> STPDE currently supports cross-domain pretraining and fine-tuning for few-shot transfer. We freeze the invariant backbone and only update the Environment Basis Manifold, allowing adaptation to a new domain with very little data. This shows the model has learned stable cross-domain patterns that can be transferred efficiently.
>
> In this sense, **STPDE has some potential as a foundation-style model**, but it does not yet achieve zero-shot forecasting. One reason is that it uses simpler temporal modeling compared with large foundation models. Improving long-term temporal modeling in future work could bring STPDE closer to zero-shot capability.
>
>
> > **Q3. Whether the Monthly-Sampled OOD Dataset Truly Exhibits Distribution Shift**
>
> In the appendix, we measured **pairwise Wasserstein distances** between monthly traffic distributions for four regions (CA-D3, SD, Orange, and LA). The results show that August consistently differs most from January, which is why we use January for training and August for testing as the OOD setting. Appendix Fig. 7 further illustrates this drift across all regions. We will clarify this more clearly in the revised manuscript.
>
> ---
>
> All supplementary experiments will be included in the Appendix. We are happy to answer any further questions.

---

> > ### Author Rebuttal · Reviewer_mv3M · 2026-04-02
> >
> > Thank you for the detailed and thorough response. My previous concerns have been satisfactorily addressed, and I will raise my score accordingly. After checking other reviews, I have one additional problem. Specifically, in ablation study, removing the stochastic perturbation module (w/o P) may result in performance decrease. This shows that stochastic perturbation is vital to enhance model performance. It would be better to provide more explanations regarding why stochastic perturbation is important.

---

> > > ### Author Response · Authors · 2026-04-02
> > >
> > > Thank you very much for your recognition of our previous response and for raising your score. We sincerely appreciate your close attention to this work.
> > >
> > > In STPDE, the Environment Basis Manifold captures local heterogeneity through sparse routing. However, when the environment representation is deterministic, the model can over-rely on environment-specific cues in the training set. Introducing stochastic perturbation—such as average embeddings or permuted bases that create structured misalignment—effectively enlarges the environment “neighborhood” seen during training. This acts as an important regularizer against non-stationary distribution shift. As a result, under severe OOD shift, the model remains more stable and robust.
> > >
> > > The experimental results support this interpretation. On the small-scale dataset CA-D3, removing perturbation (w/o P) significantly increases the error even under ID settings (+15.3%), and also degrades OOD performance (+9.1%). We believe this is because CA-D3 lacks the pattern diversity of LA; without perturbation, the model more easily overfits the limited ID data, which already harms ID performance and further weakens OOD generalization. On the large-scale dataset LA, the ID drop is smaller (+5.0%), but the error rises substantially under severe OOD shift (+15.1%). This is likely because LA contains richer spatio-temporal patterns, making overfitting less severe under ID conditions, while under strong distribution shift, the lack of perturbation-induced robustness leads to a clear OOD deterioration.
> > >
> > > We would like to thank you again for your time and valuable comments. We will carefully apply your suggestions in our work.

---

### Decision · Program_Chairs · 2026-04-30

**Decision:**

Accept (regular)

**Comment:**

This paper proposes a general framework that reformulates spatio-temporal dynamics as the evolution of inhomogeneous partial differential equations. The proposed method demonstrates good performance on in-distribution forecasting, out-of-distribution generalization, few-shot cross-city transfer, and continual learning tasks. All reviewers recognized the merits of this paper, and four of them hold positive scores for final justification. After consideration, I think this paper is well above the acceptance threshold for ICML, but I also suggest that the authors address remaining concerns about its practical advantages over simpler strong baselines.